

# Partitioning the uncertainty contributions of dependent offshore forcing conditions in the probabilistic assessment of future coastal flooding at a macrotidal site

Jeremy Rohmer[1], Deborah Idier[1], Remi Thieblemont[1], Goneri Le Cozannet[1], François Bachoc[2]

[1] BRGM, 3 av. C. Guillemin, 45060 Orléans Cedex 2, France
[2] Institut de Mathématiques de Toulouse, 118 Rte de Narbonne, 31400 Toulouse, France

*Correspondence to*: Jeremy Rohmer (j.rohmer@brgm.fr)

**Abstract.** Getting a deep insight into the role of coastal flooding drivers is of high interest for the planning of adaptation strategies for future climate conditions. Using global sensitivity analysis, we aim to measure the contributions of the offshore forcing conditions (wave/wind characteristics, still water level and sea level rise ($SLR$) projected up to 2200) to the occurrence of the flooding event (defined when the inland water volume exceeds a given threshold $Y_C$) at Gâvres town on the French Atlantic coast in a macrotidal environment. This procedure faces, however, two major difficulties, namely (1) the high computational time costs of the hydrodynamic numerical simulations; (2) the statistical dependence between the forcing conditions. By applying a Monte-Carlo-based approach combined with multivariate extreme value analysis, our study proposes a procedure to overcome both difficulties through the computation of sensitivity measures dedicated to dependent input variables (named Shapley effects) with the help of Gaussian process (GP) metamodels. On this basis, our results outline the key influence of $SLR$ over time. Its contribution rapidly increases over time until 2100 where it almost exceeds the contributions of all other uncertainties (with Shapley effect >40% considering the representative concentration pathway RCP4.5 scenario). After 2100, it continues to linearly increase up to >50%. The $SLR$ influence depends however on our modelling assumptions. Before 2100, it is strongly influenced by the digital elevation Model (DEM); with a DEM with lower topographic elevation (before the raise of dykes in some sectors), the $SLR$ effect is smaller by ~40%. This influence reduction goes in parallel with an increase in the importance of wave/wind characteristics, hence indicating how the relative effect of the flooding drivers strongly change when protective measures are adopted. By 2100, the joint role of RCP and of $Y_C$ impacts the $SLR$ influence, which is reduced by 20-30% when the mode of the $SLR$ probability distribution is high (for RCP8.5 in particular) and when $Y_C$ is low (of 50m³). Finally, by showing that these results are robust to the main uncertainties in the estimation procedure (Monte-Carlo sampling and GP error), the combined GP-Shapley effect approach proves to be a valuable tool to explore and characterize uncertainties related to compound coastal flooding under $SLR$.



## 1 Introduction

Coastal flooding is generally not caused by a unique physical driver, but by a combination of them, including mean sea-level
changes, atmospheric storm surges, tides, waves, river discharges, etc. (e.g., Chaumillon et al., 2017). The intensity of surge
itself depends on atmospheric pressure and winds as well as on the site-specific shape of shorelines and water depths
(bathymetry). Hence, compound events, resulting from the co-occurrence of two or more extreme values of these processes
is a significant reason for concern regarding adaptation. For example, flood severity are significantly increased by the co-
occurrence of extreme waves and surges at a number of major tide gauge locations (Marcos et al., 2019), of high sea-level
and high river discharge in the majority of deltas and estuaries (Ward et al., 2018), of high sea-level and rainfall at major US
cities (Wahl et al., 2015). This intensification of compound flooding is expected to be exacerbated under climate change
(Bevacqua et al., 2020). A deeper knowledge of coastal flooding drivers is thus a key element for the planning of adaptation
strategies such as engineering, sediment-based or ecosystem-based protection, accommodation, planned retreat, or avoidance
(Oppenheimer et al., 2019); see discussion by Wahl (2017).

In this study, we analyse compound coastal flooding at Gâvres town on the French Atlantic coast. This site has been
impacted by 4 major coastal flooding events since 1905 (Idier et al., 2020a); in particular, by the storm event Johanna on
March 10, 2008, which resulted in about 120 flooded houses (Gâvres mayor: personal communication; Idier et al., 2020a).
Flooding processes at this site are known to be complex (macro tidal regime and wave overtopping; variety of natural and
human coastal defences, various exposure to waves due to the complex shape of shorelines); see a thorough investigation by
Idier et al. (2020a). We aim to unravel which offshore forcing conditions among wave characteristics (significant wave
height, peak period, peak direction), wind characteristics (wind speed at 10m, wind direction) and still water level
(combination of mean sea-level, tides and atmospheric surges) drive severe compound flood events, considering projected
sea-level rise (*SLR*), up to 2200.

We adopt here a probabilistic approach to assess flood hazard, i.e. we aim to compute the probability of flooding and to
quantify the contributions of the drivers with respect to the occurrence of the flooding event by means of global sensitivity
analysis, denoted GSA (Saltelli et al., 2008). This method presents the advantage of exploring the sensitivity in a global
manner by covering all plausible scenarios for the inputs' values and by fully accounting for possible interactions between
them. The method has been applied successfully in different application cases in the context of climate change (e.g.,
Anderson et al., 2014; Wong et al., 2017; Le Cozannet et al., 2015; 2019a).

Unlike these previous studies, the application of GSA to our study site faces two main difficulties: (1) the physical processes
related to flooding are modelled with numerical simulations that have an expensive computational time cost (i.e. larger than
the simulated time). This hampers the Monte-Carlo-based procedure for estimating the sensitivity measures; (2) the offshore
forcing conditions cannot be considered independent and the probabilistic assessment should necessarily account for their
statistical dependence. This complicates the decomposition of the respective contributions of each physical drivers in GSA
(see a discussion by Do and Razavi, 2020).





Our study proposes a procedure to overcome both difficulties by combining multivariate extreme value analysis (Heffernan and Tawn, 2004) with advanced GSA techniques specifically adapted to handle dependent inputs (Iooss and Prieur, 2019) and to probabilistic assessments (Idrissi et al., 2021). To overcome the computational burden of the procedure, we adopt a metamodeling approach, i.e. we perform a statistical analysis of existing databases of pre-calculated high-fidelity simulations to construct a costless-to-evaluate statistical predictive model (named "metamodel" or "surrogate") to replace the long running hydrodynamic simulator; see e.g., Rohmer et al. (2020).

The article is organized as follows. Sect. 2 describes the test case of Gâvres, the data and the numerical hydrodynamic simulator used to assess flood hazard. In Sect. 3, we describe the overall procedure to partition the uncertainty contributions of dependent offshore forcing conditions for future coastal flooding. The procedure is then applied to Gâvres and results are analysed in Sect. 4 for future climate conditions. In Sect. 5, the influence of different scenario assumptions in addition to the offshore forcing conditions is further discussed, namely the magnitude of the flooding events, the influence of representative concentration pathway (RCP) scenario, the digital elevation model (DEM) used as input of the hydrodynamic numerical model, and the intrinsic stochastic character of the waves.

## 2 Case study

The considered case study corresponds to the coastal town of Gâvres on the French Atlantic coast in a macrotidal environment (mean spring tidal range: 4.2m). Since 1864, more than ten coastal flooding events hit Gâvres (Le Cornec et al., 2012). The flooding modelling is based on the non-hydrostatic phase-resolving model SWASH (Zijlema et al., 2011), which allows simulating wave overtopping and overflow. The implementation and validation on the study site is described in (Idier et al., 2020a), and we summarize here the main aspects. The computational domain as well as the Digital Elevation Model (DEM) are shown in Fig. 1 (red domain). The DEM (denoted DEM 2015) is representative of the 2015 local bathymetry and topography and of the 2018 coastal defences. The space and time resolution are respectively 3m in horizontal, 2 layers along the vertical dimension, and more than 10Hz. The offshore wave conditions (south of Groix island) are propagated to the boundaries of the SWASH model using the spectral wave model WW3 (Ardhuin et al., 2010) taking into account the local tide, atmospheric surge and wind (see large spatial domain in Fig. 1(a)). The combined WW3-SWASH model chain has been validated with respect to the area that was flooded during the Johanna event: the model slightly overestimates the number of flooded houses by about 3%, which can be considered very satisfactory for such complex environments.

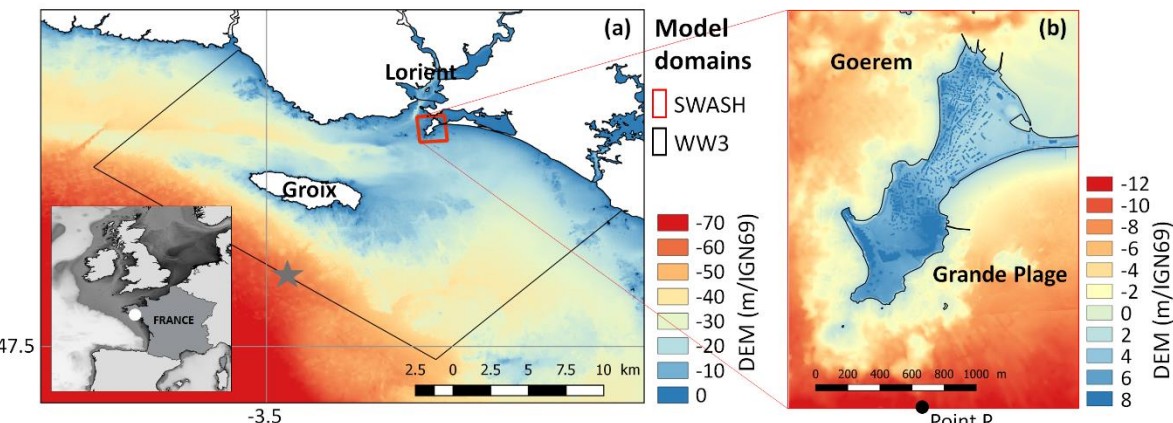

**Figure 1: Regional setting and DEM for the study site (a,b), and computational domains of the models (WW3 and SWASH) of the modelling chain (a). Adapted from Idier et al. (2020a).**

The inland impact of the storm event is numerically simulated by considering a time span of 20 minutes (with 5 minutes spin up) and steady state offshore forcing conditions. A global flooding indicator is defined as the total water volume $Y$ that has entered the territory at the end of the simulation. To account for the random character of waves, the modelling of the coastal flood induced by overtopping processes is combined with a random generation of wave characteristics in SWASH as described by Idier et al. (2020b). For given offshore forcing conditions, the simulation is repeated 20 times, and the median

value (denoted $Q_{50}$) of $Y$ is calculated as well as the quartiles (25$^{th}$ and 75$^{th}$ percentiles, respectively denoted $Q_{25}$ and $Q_{75}$). For sake of presentation conciseness, we denote also by $Y$ the median value. The impact of wave stochasticity is further discussed in Sect. 5.4. Fig. 2 provides the maps of water depth and the corresponding value of $Y$ computed with the afore-described simulator for five different storm events. In the study, we use the volume value $Y=50m^3$, $2,000m^3$ and $15,000m^3$ to categorize the flooding event as "minor", "moderate" and "very large".

The considered offshore forcing conditions are the still water level (*SWL*) – expressed with respect to the mean sea level, the significant wave height (*Hs*), the peak period (*Tp*), the direction (*Dp*), the wind speed at 10m (*U*) and wind direction (*Du*). These are defined using a database composed of hindcasts of past conditions offshore of the study site over the period 1900-2016. This dataset was built via the concatenation of hindcasts of different sources (see Idier et al., 2020a: Table 1 for further details) completed by bias corrections using a quantile-quantile correction method.


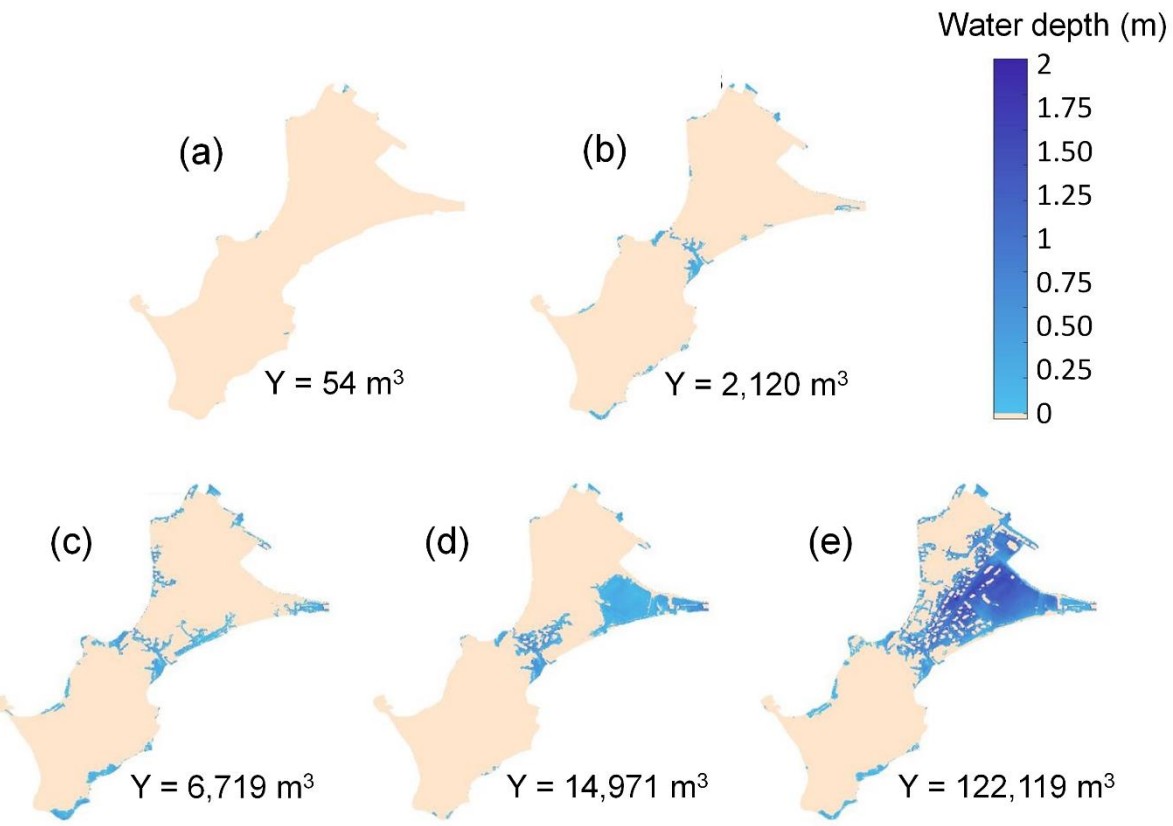

Figure 2: **Examples of five maps of water depth modelled by the numerical simulator described in Sect. 2 using DEM 2015. The value of the flood-induced inland water volume $Y$ is indicated for the five cases. In the study, the volume value 50m³, 2,000m³ and 15,000m³ have been selected to categorize the flooding event as minor, moderate and very large. Note that due to lack of numerical results with $Y$ close to 15,000 m3 in the database of simulation results (see Sect. 4.2), map (d) is provided for DEM 2008 instead.**

The uncertainty analysis is conducted for future climate conditions by accounting for *SLR* uncertainty using the probability distribution provided by Kopp et al. (2014) in the vicinity of the study site (including corrections related to vertical ground motion of -0.25 +/- 0.16 mm/y). Kopp et al. (2014)'s projections and associated uncertainty build on a combination of expert community assessment (the IPCC-AR5), expert elicitation (e.g., Bamber and Aspinall, 2013), and process modelling (e.g., the 5th phase of the Coupled Model Intercomparison Project or CMIP5) for most sea-level contributors; i.e. thermal

expansion and ocean dynamical changes, ice-sheet melting, glaciers melting and groundwater storage changes.

The data are provided with reference date 2000 for five time horizons (2030, 2050, 2100, 2150 and 2200), for 33 percentile levels and for three RCP scenarios (RCP2.6, RCP4.5 and RCP8.5). The values for intermediate time instants as well as percentile levels are obtained via interpolation (linear for percentile levels, and kriging-based (Williams and Rasmussen,





2006) for time horizons). Fig. 3 provides the projections for the three RCP scenarios as well as some examples of random

realizations following the random sampling procedure described in Sect. 3.1.

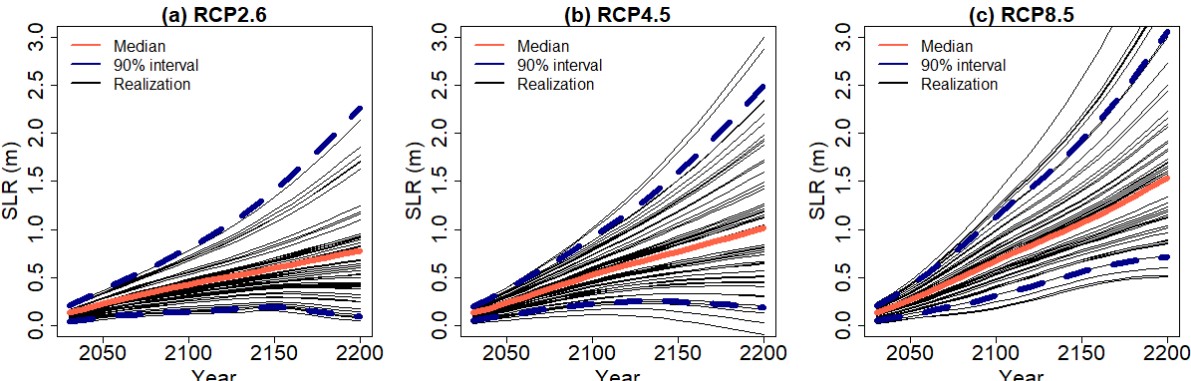

**Figure 3: Future projection of regional SLR for 3 different RCP scenarios. The red line indicates the median, and the dashed blue lines indicate the bound of the 90% confidence interval. The different black lines correspond to 50 randomly generated time series.**

## 3 Statistical methods

### 3.1 Overall procedure

First, a multivariate extreme value analysis is conducted to randomly generate a large number $N$ of "extreme-but-realistic" random realizations $\mathbf{x}$ of the scalar offshore meteo-oceanic conditions via a Monte-Carlo procedure (Sect. 3.2). The objective is to estimate the flooding probability $P_f$ defined as the probability that the inland water volume $Y$ induced by the flood

exceeds a given threshold $Y_C$, namely:

$$P_f = \text{Prob}(Y > Y_C) = \text{E}\left(\text{I}_{\{Y>Y_C\}}\right) = \text{E}(\text{I}_{\{f(\mathbf{x})>Y_C\}}) \tag{1}$$

where f(.) denotes the hydrodynamic simulator described in Sect. 2 which takes $\mathbf{x}$ as inputs to compute $Y$, E(.) is the expectation operator, and $\text{I}_{\{A\}}$ is the indicator function which takes up 1 if $A$ is true and 0 otherwise. The impact of the choice

of $Y_C$ is discussed in Sect. 5.1.

The hydrodynamic simulator f(.) has too large computation time cost (of the order of several hours per simulation depending on the input forcing conditions), which prevents its direct use for estimating $P_f$. To overcome this computational burden, we replace f(.) with a metamodel (see details in Sect. 3.3) which is trained using a limited number $n$ of pairs of inputs-output ($\mathbf{x}^i$; $y^i=f(\mathbf{x}^i))_{i=1,\dots,n}$. As metamodel, we opt for the Gaussian process (GP) regression method whose implementation and validation

are described in Sect. 3.3.

Then, using the validated GP metamodel, $P_f$ is estimated using the randomly generated realizations of the offshore conditions, and the uncertainty is decomposed into the respective contribution of the different offshore forcing conditions



using the tools of GSA (Sect. 3.4). We account for two sources of uncertainty in the estimation procedure, namely the Monte-Carlo sampling, and the GP error (related to the approximation of the true numerical model by a GP built using a finite number of simulation results) with a Monte-Carlo-based approach described in Sect. 3.5.

This GP-based GSA procedure is conducted over time by adding the sea level rise $SLR^{RCP}$ to $SWL$ for a given RCP scenario. This term is randomly sampled as follows: (1) a percentile level $SLR^{RCP}(p)$ is randomly and uniformly sampled between 0 and 1; (2) the inverse cumulative distribution function estimated from the data by Kopp et al. (2014) (see Sect. 2) is then used to sample a time series $SLR^{RCP}$ for a given RCP scenario, i.e. the same $SLR^{RCP}(p)$ level is considered over the period 2030-2200 (with a time step of 10 years); see some examples of random realizations in Fig. 3.

### 3.2 Multivariate Extreme Value Analysis

Two classes of offshore conditions are considered: 'amplitude' random variables $\mathbf{X}=(SWL, Hs, U)$, which can take up very large values, and covariates $\mathbf{X_c}=(Tp, Dp, Du)$, which are dependent on the values of the 'amplitude' variables. Considering 'amplitude' variables, a multivariate extreme value analysis (Coles, 2001) is conducted to extrapolate their joint probability density to extreme values by taking into account the dependence structure. A three-step approach is performed:

Step (1) Fitting of the marginals of 'amplitude' variables through the combination of the empirical distribution, below a suitable high threshold $u$, and of the Generalised Pareto distribution (GPD) above the selected threshold $u$ (Coles and Tawn, 1991) using the method of moments. Note that the marginal of $SWL$ is estimated by combining the marginal of the skew surge with the empirical probability distribution of tides by following the convolution approach of Simon (1994).

Step (2) The dependence structure of the 'amplitude' variables is modelled using the approach of Heffernan and Tawn (2004). Let us denote $\widetilde{\mathbf{X}}_{-i}$ the vector of all variables (with prior transformation onto common standard Gumbel margins) except the $i^{th}$ variable $X_i$. A multivariate non-linear regression model is set up as follows:

$$\widetilde{\mathbf{X}}_{-i}|\{\tilde{X}_i = \tilde{x}_0\} = \boldsymbol{a}.\tilde{x}_0 + \tilde{x}_0{}^{\boldsymbol{b}}.\boldsymbol{W}, \tag{2}$$

where $\tilde{x}_0 > \upsilon$ (i.e. $\tilde{X}_i$ having large values), $\boldsymbol{a}$ and $\boldsymbol{b}$ are parameters vectors (one value per parameter for each pair of variables), $\upsilon$ is a threshold to be defined and $\boldsymbol{W}$ is a vector of residuals. The model is adjusted using the maximum likelihood method assuming that the residuals $\boldsymbol{W}$ are Gaussian and independent of $X_i$ with a mean and variance to be calculated. Once fitted, a Monte Carlo simulation procedure is used to randomly generate realizations of the 'amplitude' variables $X$ (after transformation back on physical scales).

Step (3) Based on the generated dataset of amplitude variables, the random samples for the directional covariates $Dp$ and $Du$ are generated by using the empirical distribution conditional on the values of $Hs$ and of $U$ respectively. The peak period $Tp$ values are generated by following the approach described by Gouldby et al. (2014) based on a regression model using wave steepness conditional on $Hs$.





### 3.3 Gaussian Process Metamodel

Let us consider the set of $n$ training data $(\mathbf{x}^i; y^i = f(\mathbf{x}^i))_{i=1,\dots,n}$. In the context of GP modelling (also named as kriging, Williams and Rasmussen, 2006), we assume, prior to any numerical model run, that f(.) is a realisation of a GP ($Y(\mathbf{x})$) with

- mean (also named trend) $\mu(\mathbf{x}) = \sum_{j=1}^{k} \beta_j g_j(\mathbf{x})$ (where $g_j$ are fixed basis functions, and $\beta_j$ are the regression coefficients of the k input variables);
- stationary covariance function $k(.,.)$, (named kernel) written as $\forall \mathbf{x}, \mathbf{x}', k(\mathbf{x}, \mathbf{x}') = \mathrm{cov}\big(Y(\mathbf{x}), Y(\mathbf{x}')\big)$ with $\sigma^2$ the

variance parameter.

For new offshore forcing conditions $\mathbf{x}^*$, the predictive probability distribution $Y(\mathbf{x}^*)| \{Y(\mathbf{x}^1) = y^1, \dots, Y(\mathbf{x}^n) = y^n\}$ follows a GP with mean $m(\mathbf{x}^*)$ and variance $V(\mathbf{x}^*)$ defined using the universal kriging equations (e.g. Roustant et al., 2012) as follows:

$$m(\mathbf{x}^*) = \mathbf{g}(\mathbf{x}^*)'\hat{\boldsymbol{\beta}} + \mathbf{c}(\mathbf{x}^*)'.\mathbf{C}^{-1}.(\mathbf{y} - \mathbf{G}\hat{\boldsymbol{\beta}}), \tag{3a}$$

$$V(\mathbf{x}^*) = V_S + \big(\mathbf{g}(\mathbf{x}^*)'\hat{\boldsymbol{\beta}} - \mathbf{c}(\mathbf{x}^*)'.\mathbf{C}^{-1}.\mathbf{G}\big)'.(\mathbf{G}'.\mathbf{C}^{-1}.\mathbf{G})^{-1}.\big(\mathbf{g}(\mathbf{x}^*)'\hat{\boldsymbol{\beta}} - \mathbf{c}(\mathbf{x}^*)'.\mathbf{C}^{-1}.\mathbf{G}\big), \tag{3b}$$

where $\mathbf{C}$ is the covariance matrix between the points $Y(\mathbf{x}^1),\dots,Y(\mathbf{x}^n)$ whose element is $C[\mathrm{i},\mathrm{j}] = k(\mathbf{x}^i, \mathbf{x}^j)$; $\mathbf{c}(\mathbf{x}^*)$ is the vector composed of the covariance between $Y(\mathbf{x}^*)$ and the points $Y(\mathbf{x}^1),\dots,Y(\mathbf{x}^n)$; $\mathbf{g}(\mathbf{x}^*)$ is the vector of trend functions values at $\mathbf{x}^*$, $\mathbf{G} = (g(\mathbf{x}^1), \dots, g(\mathbf{x}^n))'$ is the experimental matrix, the best linear estimator $\hat{\boldsymbol{\beta}}$ of $\boldsymbol{\beta}$ is $(\mathbf{G}'\mathbf{C}^{-1}\mathbf{G})^{-1}\mathbf{G}'\mathbf{C}^{-1}\mathbf{y}$, and

$V_S = \sigma^2 - \mathbf{c}(\mathbf{x}^*)'.\mathbf{C}^{-1}.\mathbf{c}(\mathbf{x}^*)$ by assuming $k(.,.)$ to be stationary (Williams and Rasmussen, 2006).

To validate the assumption of replacing the true simulator f(.) by the kriging mean (Eq. 3a), we measure whether the GP model is capable of predicting "yet-unseen" input configurations, i.e. samples that have not been used for training. This can be examined by using cross-validation approaches (e.g. Hastie et al., 2009) using the following performance indicator denoted $Q^2$. Let us consider a test set $\boldsymbol{T}$ of the form $\{n+1, \dots, n+|\boldsymbol{T}|\}$ where $|\boldsymbol{T}|$ is the size of $\boldsymbol{T}$. The definition of $Q^2$ holds as

follows:

$$Q^2 = 1 - \frac{\sum_{i \in T}(y_i - \hat{y}_i)^2}{\sum_{i \in T}(y_i - \bar{y})^2}, \tag{4}$$

where $\hat{y}_i$ is the i[th] GP-based prediction of the model output $y_i$, and $\bar{y} = \frac{1}{|T|}\sum_{i \in T} y_i$ is the average value for the test set. A coefficient $Q^2$ close to 1.0 indicates that the GP model is successful in matching the new observations that have not been

used for the training.

### 3.4 Global Sensitivity Analysis and Shapley effect

Let use first focus on the presentation by considering $Y$ to be a real number (a continuous scalar variable). Among the different GSA methods (Iooss and Lemaitre, 2015), we opt for a variance-based GSA, denoted VBSA (Saltelli et al., 2008),



which aims to decompose the total variance of the variable of interest $Y$ into the respective contributions of each uncertainty;
this percentage being a measure of sensitivity.

Recall that f(.) is the hydrodynamic simulator. Consider the k-dimensional vector **x** as a random vector of random input variables $X_i$ (i=1,2,…,k) related to the different offshore forcing conditions. Then, the variable of interest $Y=f(\mathbf{x})$ is also a random variable (as a function of a random vector). VBSA determines the part of the total unconditional variance $Var(Y)$ of the output $Y$ resulting from each input random variable $X_i$. Formally, VBSA relies on the first-order Sobol' indices (ranging
between 0 and 1), which can be defined as:

$$S_i = \frac{Var\big(E(Y|X_i)\big)}{Var(Y)},$$   (5)

where E(.) is the expectation operator.

When the input variables are independent, the index $S_i$ corresponds to the main effect of $X_i$, i.e. the share of variance of $Y$ that
is due to a given $X_i$. The higher the influence of $X_i$, the lower the variance when fixing $X_i$ to a constant value, hence the closer $S_i$ to one.

When dependence/correlation exists among the input variables (as it is the case in our study), a more careful interpretation of Eq. 5 should be given: in this situation, a part of the sensitivity of all the other input variables correlated with the considered variable contributes to $S_i$, which cannot be interpreted as the proportion of variance reduction related to fixing $X_i$. To
overcome this difficulty, an extension of the Sobol' indices have been proposed in the literature, namely the Shapley effects (Owen, 2014; Iooss and Prieur, 2019; Song et al., 2016). The advantage of these effects is to allocate a percentage of the model output's variance to each input variable which includes not only the individual effect, but also the higher-order interaction and above all, the (statistical) dependence. By summing to the total variance (i.e. the sum of all normalized effects is one) and by being non-negative, the Shapley effects allow for an easy interpretation (Iooss and Prieur, 2019).
Formally, the sensitivity indices are defined based on the Shapley value (Shapley, 1953) that is used in game theory to evaluate the "fair share" of a player in a cooperative game, i.e. it is used to fairly distribute both gains and costs to multiple players working cooperatively. Formally, a k-player game with the set of players $K=\{1,2,…,k\}$ is defined as a real-valued function that maps a subset of $K$ to its corresponding cost $c: 2^K \rightarrow \mathbb{R}$ so that c($A$) is the cost that arises when the players in the subset $A$ of $K$ participate in the game. The Shapley value of player i with respect to c(.) is defined as:

$$v_i = \frac{1}{k}\sum_{A\subseteq K\backslash\{i\}} \binom{k-1}{|A|}^{-1} (c(A \cup \{i\}) - c(A)),$$   (6)

where /$A$/ is the size of the set $A$.

In the context of GSA, the set of players $K$ can be seen as the set of inputs of f(.), and c(.) can be defined so that for $A \subseteq K$, c($A$) measures the variance of $Y$ caused by the uncertainty of the inputs in $A$. Owen (2014) proposed the so-called "closed
Sobol' indices" as the cost function:



$$c(A) = S_A^{closed} = \frac{\text{Var}(E(Y|X_A))}{\text{Var}(Y)}, \qquad (7)$$

where $X_A$ is the subset of inputs selected by the indices in $A$, namely $(X_A=(X_i)_{i \in A})$.

The Shapley effect can thus be defined as:

$$Sh_i = \frac{1}{k}\sum_{A \subseteq K \setminus \{i\}} \binom{k-1}{|A|}^{-1} (S_{A \cup \{i\}}^{closed} - S_A^{closed}), \qquad (8)$$

In our study, the Shapley effect cannot be directly applied because the variable of interest $Y$ is here not a scalar, but is binary and related to the flooding probability as defined in Eq. 1. We follow the approach proposed by Idrissi et al. (2021) to extend the Shapley effect to this case (in connection with the domain of reliability assessment). In this context, we aim to investigate the influence of the inputs with respect to the occurrence of the event $\{Y > Y_C\}$, and to do so, we rely on the 'Target Shapley

effects' proposed by Idrissi et al. (2021) based on the previous definition as follows:

$$TSh_i = \frac{1}{k}\sum_{A \subseteq K \setminus \{i\}} \binom{k-1}{|A|}^{-1} (TS_{A \cup \{i\}} - TS_A), \qquad (9)$$

where $TS_A = \frac{\text{Var}(E(I_{\{Y>Y_C\}})|X_A)}{\text{Var}(I_{\{Y>Y_C\}})}$

These Target Shapley effects can be interpreted as a percentage of the variance of the indicator function allocated to the input $X_i$, and measures the influence of the input to the occurrence of the flooding event (defined by the exceedance of $Y$

above $Y_C$).

**3.5 Estimation procedure**

In practice, the Shapley effects defined in Eq. 9 are evaluated using "given data", through the post-processing of the Monte-Carlo-based results using the nearest neighbor search-based estimator developed by Broto et al. (2020) with the *sobolshap_knn* function of the R package *sensitivity*[1]. In this estimation, two major sources of uncertainty should be

accounted for, namely the Monte-Carlo sampling and the GP error (related to the approximation of the true numerical model by a GP built using a finite number of simulation results). This is done as follows:

*Step* (1) a set of *N* random realisations of the forcing conditions are generated using the methods described in Sect. 3.2;

*Step* (2) for the *N* randomly generated forcing conditions, a conditional (stochastic) *N*-dimensional simulation of the GP (knowing the training data) is generated using Eq. 3a and 3b, and the *N* values of *Y* are estimated;

*Step* (3) using the set of *N* values of *Y*, the flooding probability is estimated using Eq. 1 and the Shapley effects are computed using the nearest neighbor search-based estimator.

---

[1] https://cran.r-project.org/web/packages/sensitivity/sensitivity.pdf





Steps (1) to (3) are repeated $B$ times to generate a set of $B$ Shapley effects (one effect per forcing condition). The variability in these estimates then reflects the use of different sets of random samples (sampling error) and the use of different conditional simulations of the GP (GP error).

## 4 Application

In this section, we apply the procedure described in Sect. 3 to partition the uncertainty in the occurrence of the event $\{Y>Y_C\}$ considering the base (reference) case defined by using the median value $(Q_{50})$ of $Y$, a volume threshold $Y_C$=2,000 m³, DEM 2015 (Fig. 1), and considering the RCP4.5 scenario, i.e. for flooding event of moderate magnitude (see Fig. 2b). We select the RCP4.5 scenario because it approximately corresponds to the Intended Nationally Determined Contributions submitted in 2015 ahead of the Paris Agreement approval[2].

### 4.1 Multivariate Extreme value analysis

We use the 1900-2016 hydro-meteorological database of Idier et al. (2020a) to extract >80,000 forcing conditions. Following Step (1) described in Sect. 3.2, the extracted data are used to fit the marginals of the 'amplitude' variables using the GPD distribution with the threshold value $u$ selected using a combination of methods (visual inspection of quantile–quantile graphs, "mean residual life plots", "modified scale and shape parameters plots", see Coles et al., 2001), which yields $u_{Hs}$=6.2m, $u_{Skew\ Surge}$=0.48m, and $u_{U}$=18.9m/s corresponding to ~2 extreme events / year. The marginal distributions are provided in Appendix A.

Following Step (2) in Sect. 3.2, the dependence is modelled using the threshold $\upsilon$ of Eq. (2) set up at 0.97 (expressed as a probability of non-exceedance) using the diagnostic tools described in Heffernan and Tawn (2004: Sect. 4.4). On this basis, the Monte Carlo simulation procedure described by Heffernan and Tawn (2004) is used to randomly generate $N$=50,000 events (representative of 1,000 years). Based on the generated dataset, the corresponding covariate values are also generated (Step (3) in Sect. 3.2). Fig. 4 provides an overview of the randomly generated samples (grey dots) for the 'amplitude' variables and for the covariates. Note that some delineations (on the bottom left hand corner) can be noticed, which results from the threshold-based procedure to model the probabilistic distributions (see Sect. 3.2). This figure provides a good illustration of the moderate-to-large statistical dependence between the forcing conditions (if they were independent any structure would have been noticed), hence the need to account for it in the sensitivity analysis.

---

[2]     https://unfccc.int/process-and-meetings/the-paris-agreement/nationally-determined-contributions-ndcs/nationally-determined-contributions-ndcs



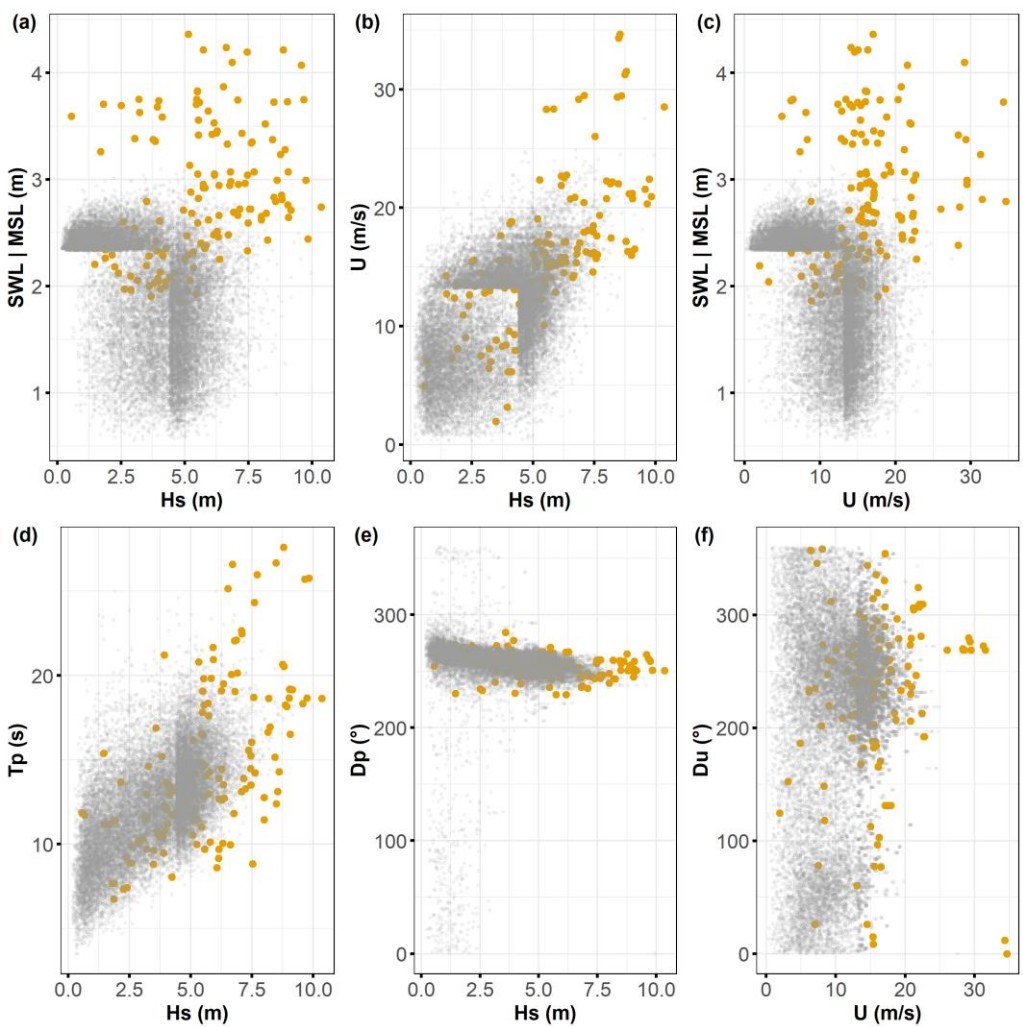

**Figure 4: Overview of the randomly generated samples using the procedure described in Sect. 3.2 (grey dots); to ease the visualization, only 20,000 samples are shown. Yellow dots correspond to the training data used to set-up the GP metamodels. They are deliberately selected outside the range of the grey dots to cover a broad range of situations (the selection approach is detailed in Sect. 4.2).**

## 4.2 GP metamodel training and validation

The objective is to train a GP model to replace the long running hydrodynamic simulator for computing $P_f$ using the $N$=50,000 randomly generated forcing conditions as described in Sect. 4.1. To do so, a limited number of simulations were run for some given forcing conditions that were selected to both cover the range of variation in Fig. 4 but also to cover very large *SWL* values. The selection is based on the approach developed by Gouldby et al. (2014) by applying clustering algorithms to a large data set of forcing conditions that are randomly generated using multivariate extreme value analysis. The reader can refer to Rohmer et al. (2020) for further details on the implementation for the site of interest here. A hundred of forcing conditions were selected via this procedure. In addition, 44 complementary cases were defined with very large




*SWL* values (see in particular the yellow dots in Fig. 4(a) –top, right). The selected forcing conditions (outlined in yellow in Fig. 4) are then used as inputs of the numerical simulations (see Sect. 2).

On this basis, a GP metamodel is trained to predict log10($Y$+1) assuming a linear trend $\mu$ and a Matérn 5/2 kernel model in Eq. 3a,b and using a maximum likelihood estimation of the GP parameters (e.g. Roustant et al., 2012). The GP metamodel is validated using a 10-fold cross validation procedure (consisting in aggregating 10 versions of Eq. 4 with varying training and

test sets): the analysis of the global performance criterion in Table 1 (1st row) also confirms this high predictive capability with $Q^2 \approx 99\%$.

**Table 1. Performance indicator of the different GP metamodels calculated using a 10-fold cross validation procedure.**

| GP model | $Q^2$ | Analysis |
|---|---|---|
| Median $Q_{50}$, DEM 2015 | 99.0 | Sect. 4 |
| 3rd quartile $Q_{75}$, DEM 2015 | 98.9 | Sect. 5.4 |
| 1st quartile $Q_{25}$, DEM 2015 | 98.9 | Sect. 5.4 |
| Median $Q_{50}$, DEM 2008 | 96.5 | Sect. 5.3 |

The scatterplot in Fig. 5 shows that the predictive capability of the trained GP model is very satisfactory (the dots almost

align along the first bisector). However, we can notice some deviations (in particular around 2.0), which justifies to account for the GP error in the GSA results by following the procedure described in Sect. 3.5.

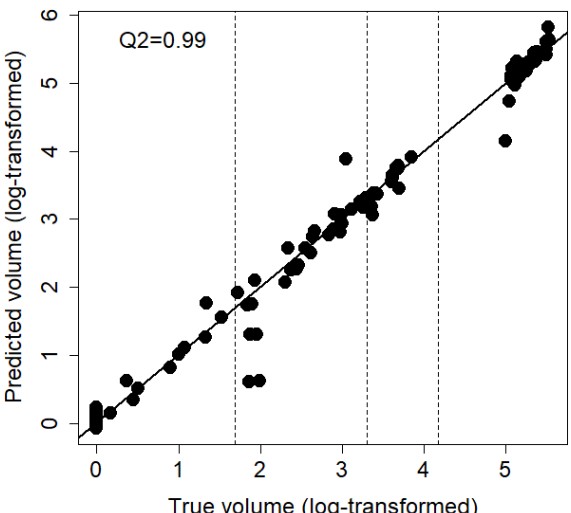

**Figure 5: Comparison between the volume estimated using the "true" numerical simulator and the ones predicted using the GP model using a 10-fold cross-validation procedure. The closer the dots to the first bisector, the more satisfactory the predictive**

**capability of the trained GP model. The performance indictor Q² is indicated and reaches here values ~99%. The vertical dashed lines indicate the threshold $Y_C$ (log-transformed) used in the study (50, 2,000 and 15,000 m³).**

### 4.3 Uncertainty partitioning over time

The $N$=50,000 randomly generated forcing conditions in addition to the random *SLR* time series (see some examples in Fig. 3) are used as inputs of the validated GP models to evaluate the time evolution of $P_f$ for the base case (Fig. 6). Preliminary

convergence analysis showed that 50,000 Monte-Carlo samples were sufficient to reach stable results; this is also shown by the very small uncertainty band's width in Fig. 6 (see in particular the inserted plot) defined by the lower and upper bounds computed using 50 replicates of the estimation procedure (described in Sect. 3.5). This also shows that both error sources (GP and sampling) have minor influence.

Fig. 6 shows that $P_f$ increases over time in a non-linear manner and reaches values of 10% in the long term, by 2100 and

~44% in the very long term, by 2200, i.e. equivalent to a return period (inverse of $P_f$) of respectively 10 years and about 2.3 years.

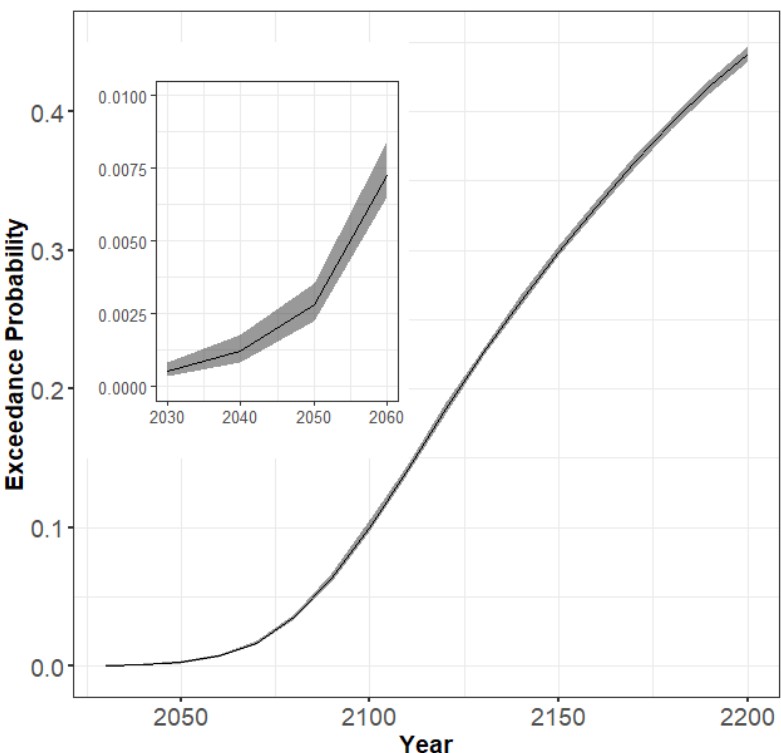

**Figure 6: Time evolution of the flooding probability for the base case so that the median value (denoted $Q_{50}$) of the inland water volume induced by the flood exceeds the threshold of $Y_C$=2,000m³ given RCP4.5 and DEM2015. The inserted figure indicates the**
**very small uncertainty band's width whose limits are the lower and upper bounds computed using 50 replicates of the estimation procedure (Sect. 3.5) accounting for GP and sampling error.**

The Shapley effects for the flooding event {$Y$>2,000m3} were evaluated using the 50,000 GP model evaluations using the procedure described in Sect. 3.5 and the nearest neighbor search-based estimator (with 5 neighbors and a pre-whitening of the inputs with the ZCA-cor procedure, Kessy et al., 2018). Preliminary convergence analysis showed us that 50,000 samples





were sufficient to reach low uncertainty estimates as shown by the error-bars provided in Fig. 9. This also confirms that both error sources (GP and sampling) have small influence. Fig. 7 depicts the time evolution of the Shapley effects, which measure the influence of the inputs on the occurrence of the flooding event.

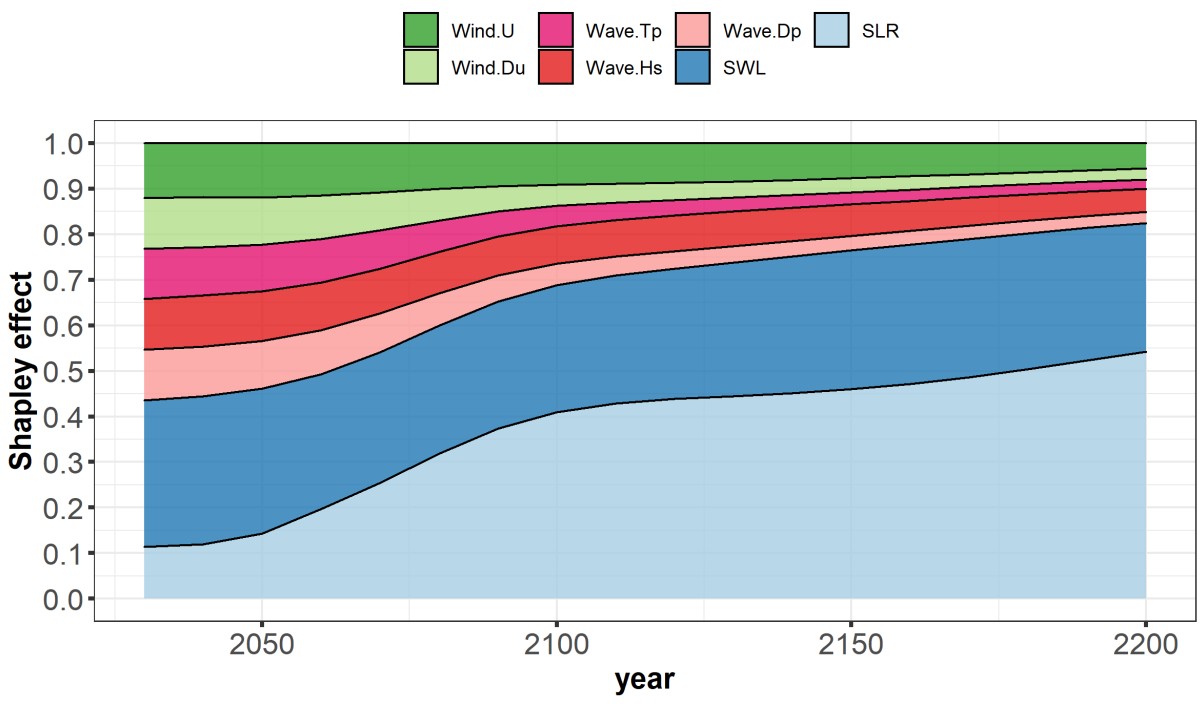

**Figure 7: Time evolution of the Shapley effects for the base case (RCP4.5, $Y_C$=2,000m3, DEM 2015 and use of the median value to**
**account for wave stochasticity) estimated by computing the median value from 50 replicates of the estimation procedure (Sect. 3.5)**
**accounting for GP and sampling error.**

Several effects can be seen:

- The influence of *SLR* increases over time with a non-negligible contribution of ~15% even in the short term (<2050) until reaching ~40% in the long term (2100) by following a relatively steep evolution (with an increase by
180% from 2020 to 2100);

- After 2100, *SLR* contribution continues to increase until reaching ~55% in the very long term (2200) but by following a relatively gentle linear evolution (with an increase by 37.5% over a 100-year time duration from 2100 to 2200). This means that by 2200, *SLR* dominates the cumulative contributions of all remaining uncertainties;

- In the short term, the major contributor corresponds to *SWL*. The Shapley effect is of ~32%, while the remaining
forcing conditions have contributions of the order of 10-13%. We also note that by 2080, *SLR* Shapley effect exceeds the one of *SWL*;





- Over time, the contributions of all forcing conditions (except *SLR*) decrease (to compensate the *SLR* increase because the sum of all Shapley effects is one) until reaching a quasi-horizontal plateau by 2100. The Shapley effects are of the order of 28% for *SWL* and 8-9% for *H*s and *U*, hence indicating their small-to-moderate influence though non-negligible;


- The Shapley effects of the covariates (*D*p, *T*p and *D*u) reach however low values <3-4%, which provides a strong evidence of their negligible role for time horizon >2100, i.e. their individual effect as well as their dependence and their interactions with the other variables are almost zero.

## 5 Influence of the scenario assumptions

Here, we investigate to which extent alternative scenario assumptions underlying our approach might change the afore-described conclusions, namely: the volume threshold $Y_C$ used to define when a flooding event occurs (Sect. 5.1); the RCP scenario (Sect. 5.2); the use of an alternative DEM (Sect. 5.3); and the impact of wave stochasticity (Sect. 5.4). For each analysis, the corresponding assumption was changed and the whole analysis (described in Sect. 3.1) was re-conducted, i.e. (1) new hydrodynamic simulations; (2) training of new GP models (the predictive capability is provided in Table 1); (3) GP-

based estimate of the flooding probability and of the Shapley effects within a Monte-Carlo-based simulation procedure (Sect. 3.5).

**Figure 8: Time evolution of the flooding probability for different assumptions: volume threshold $Y_C$ (a); RCP scenarios (b); DEM (c); and wave stochasticity modelled with different quartiles (respectively denoted $Q_{25}$, $Q_{50}$ and $Q_{75}$) (d). The inserted figures indicate the small uncertainty band's width estimated with the lower and upper bound computed for 50 replicates of the estimation approach (Sect. 3.5) accounting for GP and sampling error. The lines correspond to the median value computed for 50 replicates. The green lines represent the base case (defined for the median value $Q_{50}$ of $Y$ with a threshold of $Y_C$=2,000m³ given RCP4.5 and the DEM 2015). Note that the different flooding probability estimates for (d) overlap due to minor influence of wave stochasticity.**





## 5.1 Volume threshold

The uncertainty partitioning in Sect. 4 was performed given a threshold $Y_C$=2,000m3 that determines whether a flooding event occurs or not. The analysis is here re-conducted by respectively focusing on minor and very large flooding events defined for $Y_C$=50 and 15,000m$^3$ (as illustrated in Fig. 2a and 2c respectively) considering the other assumptions of the base case.


Fig. 8(a) shows the time evolution of the flooding probability for the three cases. Compared to the base case (outlined in green), the results indicate that the occurrence of the minor (respectively very large) flooding events have a higher (lower) probability by 19% (~4%) in the long term (2100). The time evolution of the uncertainty contribution is modified depending on the chosen threshold (Fig. 9). When $Y_C$=50m$^3$, *SWL* has the higher contribution (of 44%) in the short term (compare the

blue with the green and orange bars in Fig. 9a), while the other offshore conditions all have lower influence (with Shapley effects lower by 2-~5%) compared to higher $Y_C$ values. This is however not the case for *SLR* which has approximately the same influence whatever $Y_C$ in the short term. Over time (Fig. 9b,c), the case $Y_C$=50m$^3$ presents a slower increase of *SLR* influence, which is directly translated into a slower reduction of *SWL* contribution (compare the two first groups of bars in Fig. 9c).

This *SLR*-threshold relation directly reflects how *SLR* acts on the flooding likelihood: it acts as an "offset", which means that it induces a higher sea water level at the coast; hence a higher likelihood of flooding. Thus, the lower the threshold value, the lower the necessary *SLR* magnitude to induce flooding, hence the lower influence.

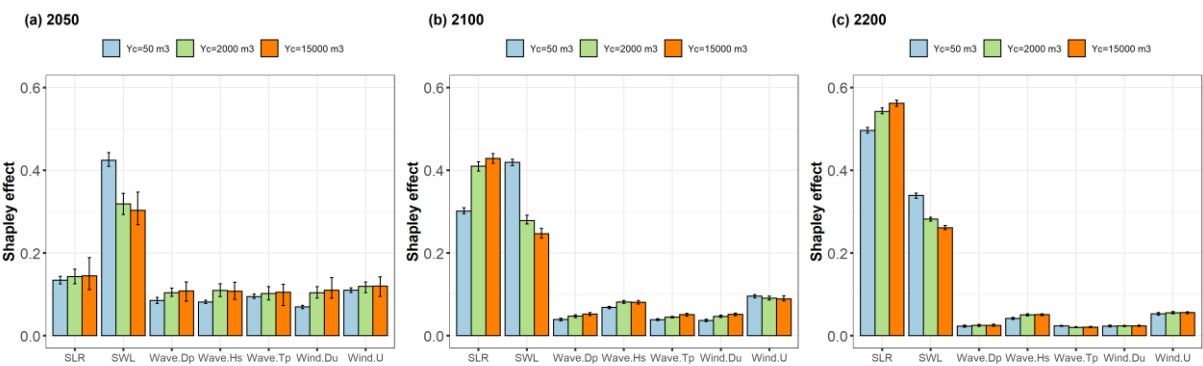

**Figure 9: Shapley effects for different volume threshold values $Y_C$ at three time horizons: in the short term, by 2050 (a); in the long term by 2100 (b) and in the very long term by 2200 (c). The green bars correspond to the base case analysed in Sect. 4. The bar height is estimated by the median value over 50 replicates of the Monte-Carlo-based estimation approach (Sect. 3.5) accounting for GP and sampling error. The lower and upper bounds of the error bar are estimated by using the 50 replicates.**

## 5.2 RCP scenario

The uncertainty partitioning in Sect. 4 was conducted given the RCP4.5 scenario i.e. given a scenario related to relatively moderate *SLR* magnitude (Fig. 2b), compared to RCP8.5 in particular (Fig. 2c). The analysis is here re-conducted for the two



other RCP scenarios. Compared to the base case, Fig. 8(b) indicates that the occurrence of the flooding events for RCP8.5 (respectively RCP2.6) have a higher (lower) probability by about 10% (-5%) for time horizon 2100. Regarding the time evolution of the Shapley effects, it appears to be the same regardless of the scenario in the short term (Fig. 10a), but starts to

differ in the long term (Fig. 10b) and even more significantly in the very long term (Fig. 10c). We note that for RCP8.5, the influence of *SLR* is lower by almost 20% relative to RCP2.6, which is translated into a higher influence of *SWL*.

Like in Sect. 5.1, this is the "offset" effect of *SLR* that influences the most: for RCP8.5, the mode of the *SLR* distribution (in red in Fig. 2c) exceeds the one of the other scenarios after 2100, and can induce a high water level at the coast, hence potentially a water volume value close to $Y_C$=2,000 m$^3$, and a higher flooding likelihood. This means that *SLR* values

sampled around the mode will less impact the occurrence of the flooding event (and the flooding probability), because a small *SLR* offset is here necessary to trigger the flooding event. This is not the case for the two remaining scenarios, because the mode is of lower magnitude and any *SLR* values sampled above it will have a key impact on the flooding occurrence.

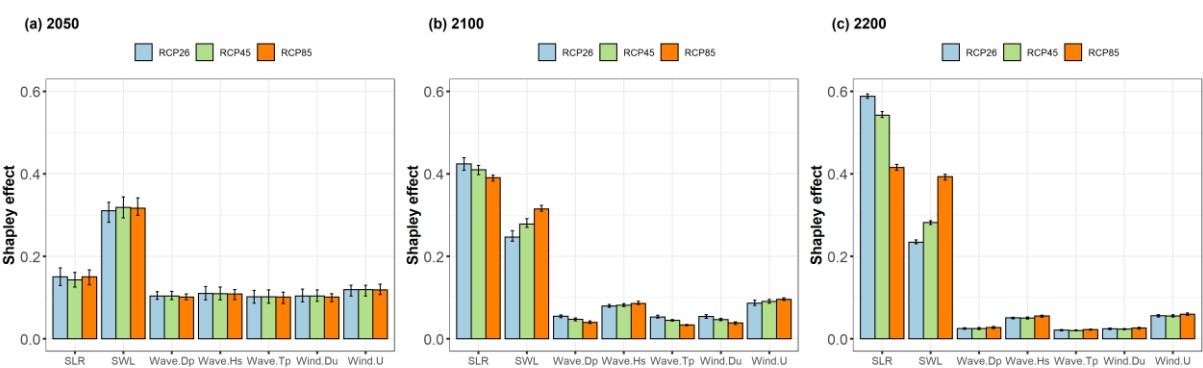

**Figure 10: Shapley effects for different RCP scenarios at three time horizons: in the short term 2050 (a); in the long term by 2100 (b) and in the very long term by 2200 (c). The green bars correspond to the base case analysed in Sect. 4. The bar height is estimated with the median value calculated with 50 replicates of the estimation approach (Sect. 3.5) accounting for GP and sampling error. The lower and upper bounds of the error bar are estimated using the 50 replicates.**

### 5.3 DEM

The flooding simulations described in Sect. 2 were performed using a DEM that is representative of the conditions of 2015. Here, we investigate to which extent an alternative DEM might change the sensitivity analysis results. To do so, we use the DEM 2008 (with the same resolution of 3m than DEM 2015), which corresponds to the conditions before the major flooding event of Johanna 2008  i.e. prior to the protectives measures relying on the raise of the dykes in the aftermath of this event. This means that DEM 2008 presents some sectors of lower topographic elevation of coastal defences (see Appendix B: Fig.

B1) compared to DEM 2015 (in particular on the south-eastern sector, which is highly exposed to storm impacts; see Appendix B: Fig. B1(b)).

Fig. 8(c) shows the time evolution of $P_f$ for both DEMs: this indicates that the changes in topography leads to small-to-moderate differences in $P_f$ (compared to the changes resulting in the other scenario assumptions as outlined in Fig. 9a,b) with



a lower value for DEM 2015 before 2100 (with a maximum difference of ~3%), and a reverse tendency after 2100 (with a
slightly lower order of magnitude of ~2%). These differences in the short/long term were expected due to the lower elevation
in some sectors of DEM 2008 (Appendix B), but they remain more difficult to explain in the very long term. An extraction
of the forcing conditions (Fig. 11) that lead to exceeding $Y_C$=2,000 m$^3$ by 2200 for DEM 2015 but not for DEM 2008,
reveals that the small differences in $P_f$ is related to very particular storm events with high $SWL$, moderate-to-high $SLR$ level,
small-to-moderate wave and wind contributions and very specific wave direction with mode at ~270°. This explorative
analysis points out the importance of the forcing conditions in the occurrence of the flooding, which is further analysed with
the Shapley effects in Fig. 12.

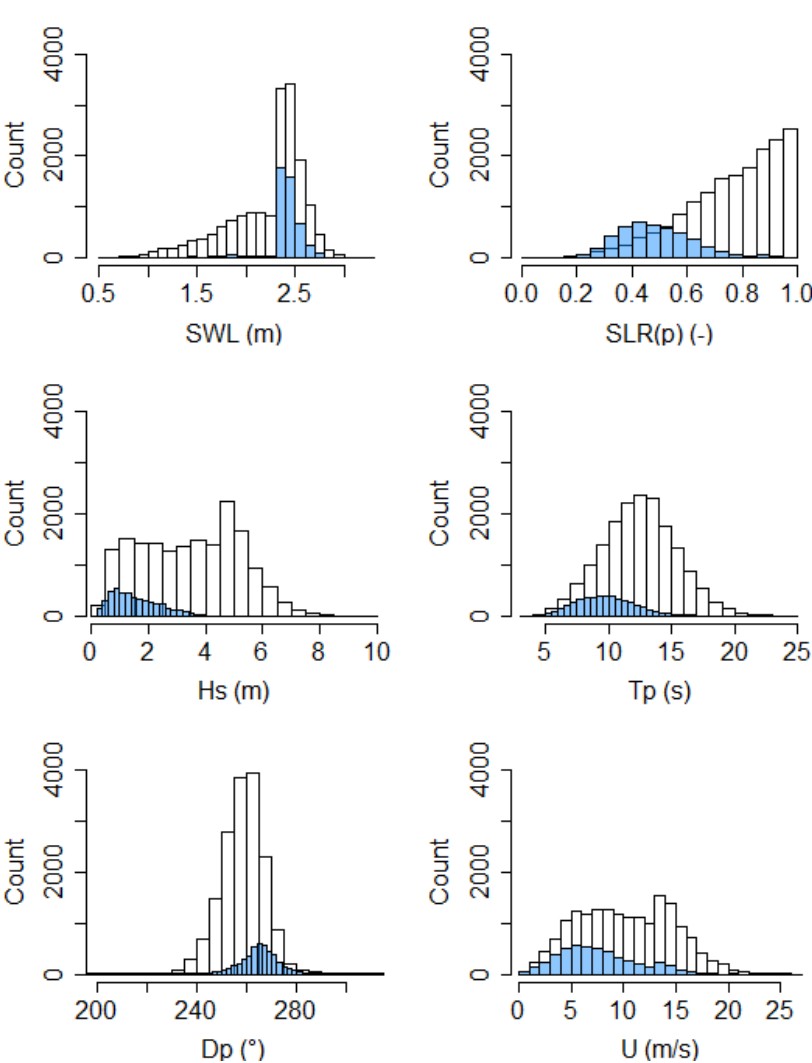

**Figure 11: Histogram calculated for 50,000 random forcing conditions that lead to exceeding $Y_C$=2,000m3 by 2200 for both DEMs
(DEM 2008 and 2015, white) and for DEM 2015 but not for DEM 2008 (light blue). $SLR$(p) denotes the percentile level of the $SLR$
time series (see Sect. 3.1). The wind direction is not plotted because of the very minor changes between both histograms.**





Whatever the time horizon, Fig. 12a,b shows that the influence of wave characteristics, *Hs* and *T*p, is higher for DEM 2008 by up to ~10% (as well as *D*p but to a lesser extent), while the influence of wind characteristics (*U* and *Du*) is lower. There are also differences in the influence of *SLR* with a lower contribution for DEM 2008 (compensated by a higher influence of *SWL*). These differences increase over time until reaching values up to 20% (Fig. 12b), then vanish after 2100 in the long

term (Fig. 12c). These results relate to the different drivers of flooding depending on the DEM: the sectors with lower topographic elevation have a higher sensitivity to wave-induced flooding, i.e. overtopping at least until 2100. In the very long term, both cases present a low influence of wave characteristics, hence indicating a lower contribution of overtopping. This may be explained by the changes in the breaking process of waves in relation to large *SLR* values. In particular, the potential reduction in overtopping in relation to lower likelihood of impulsive waves was shown by Hames and Gouldby

(2021). But this study was restricted to vertical sea walls, and needs to be extended to cover the large spectrum of different types of coastal defences (artificial, sand dunes) in our case study.

Our analysis illustrates how adaptation measures mainly based on raising dykes might affect the drivers of the different flooding processes (overtopping, overflow or combined), but also its complexity. Though all physical processes are not fully understood here (in particular in the very long term, >2100), our results justify deepening the analysis through

complementary investigations for instance based on numerical simulations.

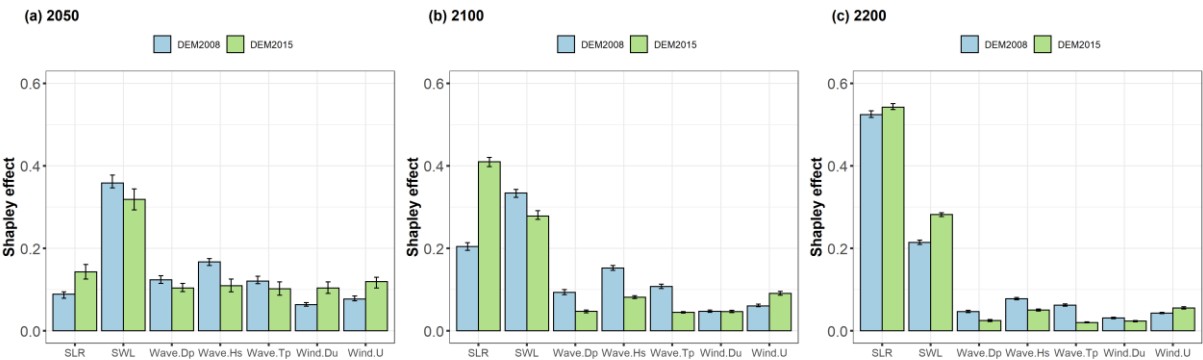

**Figure 12: Shapley effects for two different DEMs at three time horizons: in the short term 2050 (a); in the long term by 2100 (b) and in the very long term by 2200 (c). The green bars correspond to the base case analysed in Sect. 4. The bar height is estimated**
**with the median value calculated with 50 replicates of the estimation approach (Sect. 3.5) accounting for GP and sampling error. The lower and upper bounds of the error bar are estimated using the 50 replicates.**

**5.4 Wave stochasticity**

The effect of wave stochasticity was tested by re-conducting the whole analysis but using the $1^{st}$ quartile (denoted $Q_{25}$), or the $3^{rd}$ quartile (denoted $Q_{75}$) instead of the median (denoted $Q_{50}$) of *Y*. Figure 8(d) confirms the very minor influence of

wave stochasticity on the flooding probability estimate (with differences <1%). The uncertainty partitioning is also very little influenced (Fig. 13) whatever the time horizon. This result differs from the one of Idier et al. (2020b), who showed the importance of this effect that was there comparable to the one of *SLR* as long as the still water level remains smaller than the





critical level above which overflow occurs. The differences between both studies may be explained by the differences in the procedure. Idier et al. (2020b) analysed this effect for two specific past storm events, whereas our study covers a large

number of events by randomly sampling different offshore forcing conditions (Fig. 4). To conclude on this effect (relative to the others), further investigations are thus necessary and could benefit for instance from recent GSA for stochastic simulators (Zhu and Sudret, 2021).

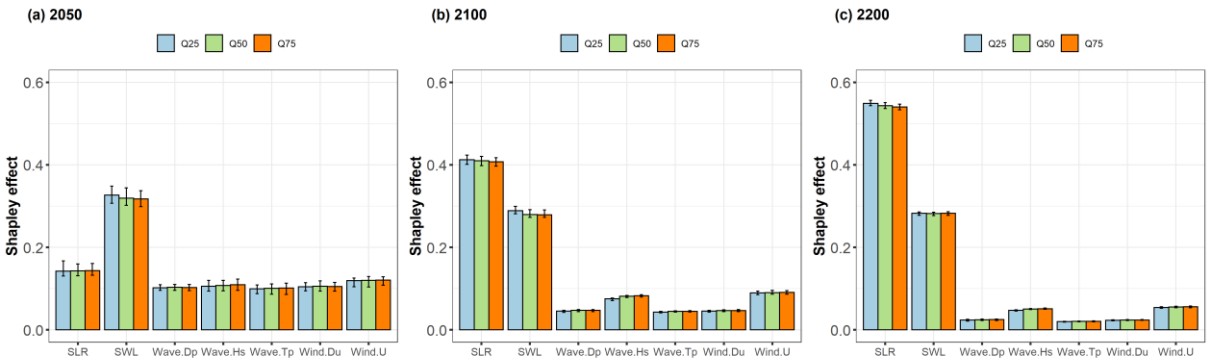

**Figure 13: Shapley effects using quartiles (denoted Q$_{25}$,Q$_{50}$,Q$_{75}$) of the water volume at three time horizons: in the short term 2050 (a); in the long term by 2100 (b) and in the very long term by 2200 (c). The green bars correspond to the base case analysed in Sect. 4. The bar height is estimated with the median value calculated with 50 replicates of the estimation approach (Sect. 3.5) accounting for GP and sampling error. The lower and upper bounds of the error bar are estimated using the 50 replicates.**

## 6 Summary and further works

At the macrotidal site of Gâvres (French Britany), we have estimated the time evolution of the flooding probability defined so that the inland water volume $Y$ induced by the flooding exceeds a given threshold $Y_C$. For moderate flooding events (with $Y_C$=2,000m3), the flooding probability rapidly reaches ~10% (return period of 10 years) by 2100 and (quasi-)linearly increases until ~44% (~2.3years) in the very long term (by 2200). The minor (with $Y_C$=50m$^3$), respectively very large (with $Y_C$=15,000m$^3$), flooding events have a higher (lower) probability of 19% (~4%) in the long term (2100), i.e. equivalent to a

decrease (increase) of the return period from 10 to ~5 years (25 years).

By relying on Shapley effects, our study underlines the key influence of $SLR$ on the occurrence of the event $\{Y > Y_C\}$ regardless of $Y_C$ value. This influence (with a non-negligible Shapley effect of 15% in our base case) starts early, i.e. from the short term for a time horizon <2050, then rapidly increases over time until 2100 for which it almost exceeds the contributions of all other uncertainties, and after 2100, it continues to linearly increase up to values >50%. The $SLR$

influence is almost dual of the one of $SWL$, i.e. the increase in $SLR$ Shapley effects is almost compensated by a decrease of the same order of magnitude for $SWL$ Shapley effects (due to the constraint that the sum of all Shapley effects is one).

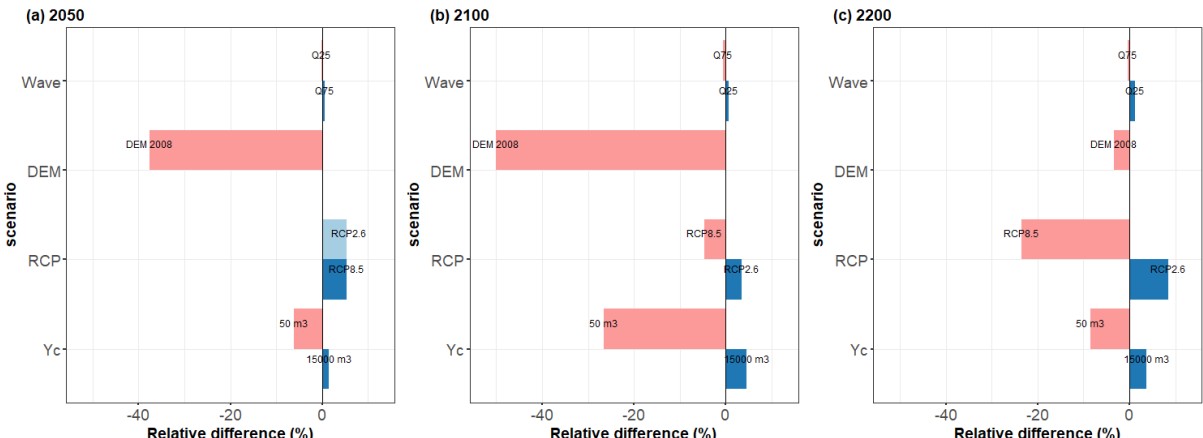

**Figure 14: Relative differences of the Shapley effect for *SLR* (using the median values computed for 50 replicates of the estimation procedure) depending on the scenario assumptions (volume threshold $Y_C$, RCP scenario, DEM and wave stochasticity) with respect to the base case at different time horizons (2050 (a), 2100 (b) and 2200 (c)).**

However, the *SLR* effect is shown to both depend on the scenario assumptions and on the considered time horizon. Before 2100, it is strongly influenced by the DEM (Fig. 14a,b); with a DEM with lower topographic elevation, the *SLR* effect is smaller compared to the base case and higher contributions of wave characteristics are shown by our analysis. By 2100, the threshold and the RCP scenario play a joint role: the higher the mode of the *SLR* probability distribution (for RCP8.5 scenario in particular) and the lower the $Y_C$ value, the lower the *SLR* influence (with respect to the base case, Fig. 14b,c).

The contribution of wave and of wind characteristics remains of the order of 5-15% (for the base case), and reduces over time until reaching in the long term low values, based on which the covariates, $D_p$, $D_u$ and $T_p$ can be considered negligible. This result would not have been shown if 'traditional' sensitivity analysis (using Sobol' indices) had been used, because the strong dependence among the inputs would not have been accounted for (Appendix C). Finally, we note that the sensitivity analysis results are very little influenced by wave stochasticity in our case.

The analysis of the main uncertainties in the estimation procedure (Monte-Carlo sampling and GP error) shows here minor impact, which is a strong indication that the combined GP-Shapley effect approach is a robust tool worth integrating in the toolbox of coastal engineers and managers to explore and characterize uncertainties related to compound coastal flooding under *SLR*. However, to reach an operative level, several aspects should be improved.

First, regarding the input data, we used the *SLR* projections from Kopp et al. (2014). These are generally consistent with the latest IPCC sea-level projections presented in the Special Report on Ocean and Cryosphere in a Changing Climate (Oppenheimer et al., 2019). Yet, the highest quantiles may not represent well the possibility of marine ice-sheet collapse in Antarctica (De Conto et al., 2021). The lowest quantiles of the Kopp et al. (2014) projections need to be considered even more cautiously, the 17% quantile being a reasonable low-end scenario given the scientific evidence available today (Le Cozannet et al., 2019b). The study should be updated using the new projections in the future.
Second, from a methodological perspective, we investigated the uncertainty in the sensitivity analysis results by estimating the influence of the scenario assumptions with respect to a base case by relying on a parametric analysis. A more general framework is here needed to incorporate different levels of uncertainty, i.e. a first level that corresponds to the forcing conditions, a second level that is related to the scenario assumptions (choice in RCP, DEM, in the threshold value, etc.) and a
third level that is related to the stochastic nature of our numerical model. This 'sensitivity analysis of sensitivity analysis' deserves further research in the future (as outlined by Razavi et al., 2021: Sect. 3.5).

**Author contributions**

JR, DI and GLC designed the concept. JR, DI and FB set up the methods. JR, DI, RT, GLC, and FB set up the numerical experiments. DI performed the numerical analyses with the hydrodynamic model. RT and GLC pre-processed and provided
the projection data. JR undertook the statistical analyses. JR wrote the manuscript draft, DI, RT, GLC and FB reviewed and edited the manuscript.

**Competing interests**

The authors declare that they have no conflict of interest.

**Code/Data availability**

Code are available upon request to the first author.

**Acknowledgements**

The authors thank the ANR for its financial support to the RISCOPE project (ANR-16-CE04-0011). We thank F. Gamboa (IMT), T. Klein (IMT/ENAC), and R. Pedreros (BRGM) for fruitful discussions.





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





**Appendix A Extreme value analysis - Marginals**

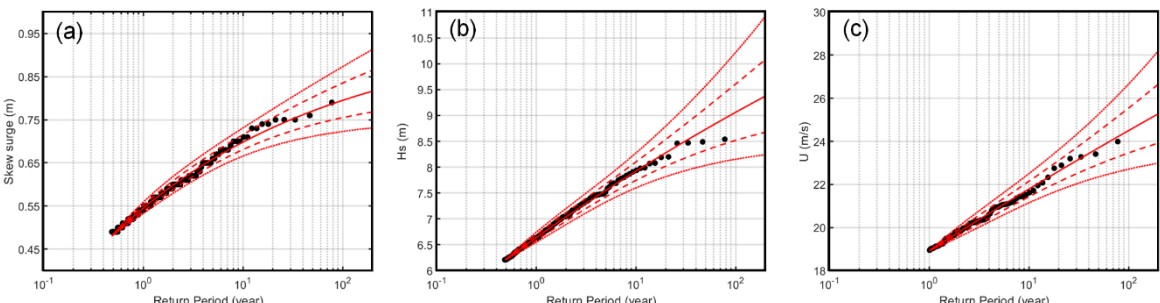

**Figure A1: Marginal distributions for the 'amplitude' variables: skew surge (a), *H*s (b), *U* (c). Black dots represent the empirical data (using Hazen formula for the return period). The red line represents the best fit of the GDP distribution. The dashed and dotted lines respectively represent the bootstrap-based confidence intervals at 70 and 95%.**


**Appendix B Differences between DEM 2015 and 2008**

**Figure B1: DEM 2015 (a); Difference (denoted DZ) between DEM 2015 and DEM 2008 (b); Elevation for DEM 2015 and 2008 along transects P1 (c) and P2 (d).**



**Appendix C GSA with first-order Sobol' indices**

Figure C1 provides the GSA results using the first-order Sobol' indices (which do not account for the dependence among the variables, see Sect. 3.4). This shows that the identification of the major contributors to the uncertainty would have been similar, i.e. high influence of *SWL* in the short term, and an increase impact of *SLR* until reaching an overwhelming role in the very long term. Yet, the interpretation of the importance of the remaining variables would have been more difficult, if not leading to wrong conclusions. Before 2100, we note very low values of the corresponding 1st order Sobol' indices from

which a low influence could have been concluded. Yet, these indices account neither for the interaction terms (measured by one minus the sum of all 1st order indices as outlined by the large white area in Fig. C1) nor for the dependence (as clearly shown in Fig. 4). Here, the Shapley effects are useful to clearly indicate the significant (though moderate) importance of these variables, and more importantly they provide a strong evidence that the covariates *Dp*, *Du* and *Tp* have all minor influence in the very long term because of their low Shapley effect values.

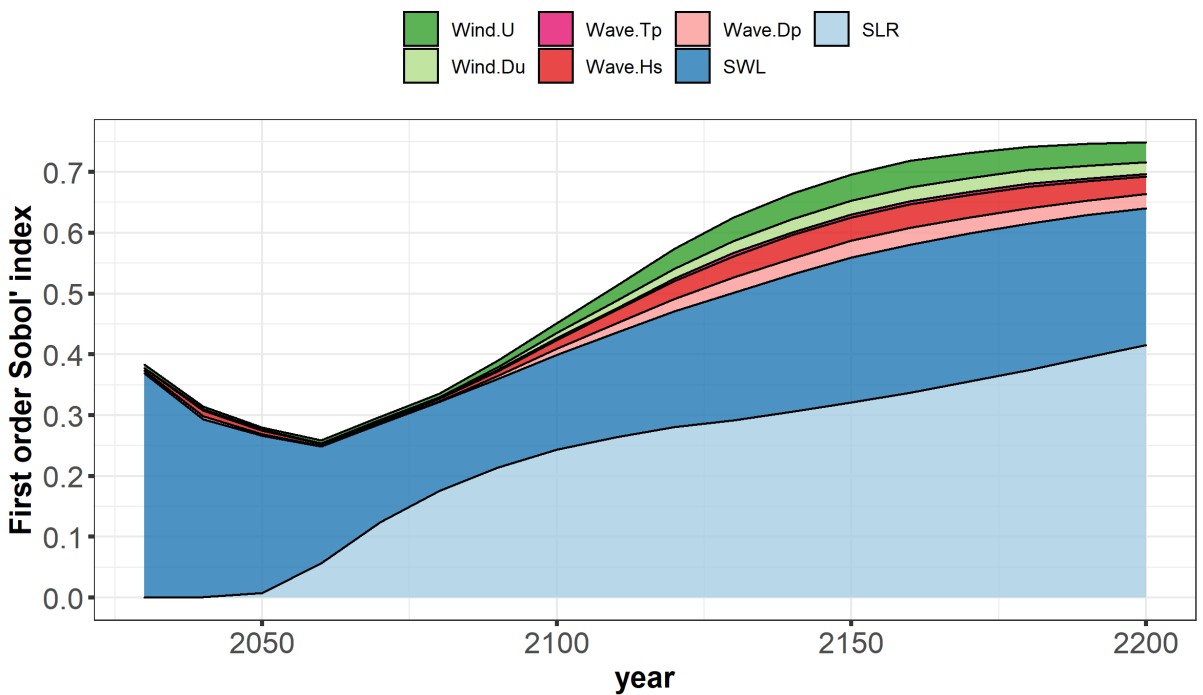


**Figure C1: Time evolution of the first-order Sobol' indices for the base case (RCP4.5, $Y_C$=2,000m3, DEM 2015 and use of the median value to account for wave stochasticity) estimated by computing the median value from 50 replicates of the estimation procedure (Sect. 3.5) accounting for GP and sampling error.**