# Peer review of "Partitioning the contributions of dependent offshore forcing conditions in the probabilistic assessment of future coastal flooding"

_Natural Hazards and Earth System Sciences, 2021_

## Referee Comment (RC1)

**Partitioning the uncertainty contributions of dependent offshore forcing conditions in the probabilistic assessment of future coastal flooding at a macrotidal site.**

**General comment**

The authors present an interesting methodological framework to assess the uncertainty contributions of forcing conditions to the probability of future flooding. Then, the framework is applied to a macrotidal study site at the French Atlantic coast, and its sensitivity to different framework assumptions is also tested.

The paper is quite interesting both in terms of proposed methods and yielded results, and will be of interest for NHESS readers.

The methodological framework is complex, and because of the way the paper is structured, some parts of the method are under the Section 4 (Application) instead of Section 3 (Methods). As a general comment, I would recommend a slight review of the current structure of the manuscript, trying to wrap all methodological information in section 3, leaving sections 4 and 5 to purely present results. A methodological chart might also help in such a complex study.

In line with the previous comment, current section 6 presents a summary of results and future works. I would also recommend splitting this in Discussion and Conclusions. The results summary is repetitive and the manuscript could be improved with a broader discussion comparing the used methods (e.g. the Heffernan and Tawn 2004 approach, or the GPs) with other methods available in literature such as hierarchical copulas or RBFs (e.g. in Goulby et al., 2014). Some discussion on limitations is done in current chapter 6 although it seems short for such a complex study such as the one performed here, with many methodological steps.

**Specific comments**

Line 33: "…flood severity _is_ significantly increased..

Line 112: Maybe the sentence is lacking the verb: "..were built on…".

Line 266-270: This is general methodology. Consider defining the base case scenario and the alternative scenarios to test sensitivity on different assumptions under the methodological sections.

Line 304: What does _aggregating_ mean in this context? From what I understand, results of $Q^2$ in Table 1 should be given as a _mean +- std_ as they are the result of a 10-fold cross validation procedure. How is the single value given in table 1 computed from the 10 folds?

Figure 5: is it showing 1 fold or the 10 folds? A word on the effect that the poorer performance of the GP approach around $Yc = 50 \text{ m}^3$ might have on the results might be interesting in the discussion.

Line 332-333: the ZCA-cor procedure and number of neighbors are both parameters of the R function, namely _n.knn_ and _rescale_. This part and associated reference can be moved to the methods section, as it is confusing here.

Line 335. It is not standard to present Figure 9 before Figures 7 and 8. Consider moving this part later in the manuscript.

Line 360-366: As in Line 266-270, this is general methodology. Consider defining the base case scenario and the alternative scenarios to test sensitivity on different assumptions under the methodological sections.

Line 428. Do you men Figure 8a,b?

---

## Referee Comment (RC3)

**General comments:**

The manuscript "Partitioning the uncertainty contributions of dependent offshore forcing conditions in the probabilistic assessment of future coastal flooding at a macrotidal site" by Rohmer et al. presents an interesting framework to assess uncertainties in future flood risk projections at a site in France, which undoubtedly falls under the scope of NHESS.

While it is clear that a lot of work has gone in this study and manuscript preparation, there are some improvements that are needed before the manuscript should be accepted for publication. These should include some clarifications with respect to the methodology, an update in the structure of the paper and some language improvements. The methodology and results are quite interesting, but due to the complexity of the paper, the current structure is not optimal. For example, Section 4 named "Application" has a mixture of methods and results. Subsections 4.1 and 4.2 could be incorporated in section 3. Sections 2 and 3 could be merged and reorganized as well, starting with a presentation of the study site and available data, then the multivariate analysis, then describing the hydrodynamic model, then the GP meta-model and so on. A nice diagram with the methodological framework and references to the respective sections could really help the reader. This could be placed at the end of the introduction sections or in the beginning of the methodological section. Moreover, the figures are a lot and some of them (Figures 9-13) could go in the Appendix without any problem with the flow of the paper.

While the dependency of the offshore parameters seems to be one of the main highlights of the paper, the presentation of this part and especially Figure 4, are not that clear. Some extra dependency indicators and improvements in the figure could help to clarify this (see my comments bellow for more specific information). Moreover, I would expect that the duration of an offshore event is an important parameter when assessing flood risk. However, the authors use a fixed duration of 20 minutes with uniform conditions. I would appreciate if the authors justify this choice.

Additionally, I feel that for such a complex methodological framework the discussion (including limitations sections) is rather short. Maybe section 5 could actually go in the discussion section (I feel like it can belong there since it discusses the assumptions used), especially if the figures that accompany it, are moved to the appendix. Other limitations and assumptions should be discussed as well; the way the dependency structure is modelled relatively to other methods available like e.g., copulas; the use of GP as a meta-model versus other statistical techniques etc.

In general, the language could improve as well, as I noticed there were some grammar issues and typos here and there.

**Specific comments:**

Line 54: Athanasiou et al. 2020 applied GSA as well for coastal erosion projections at the European scale

Athanasiou, P., van Dongeren, A., Giardino, A., Vousdoukas, M.I., Ranasinghe, R., Kwadijk, J., 2020. Uncertainties in projections of sandy beach erosion due to sea level rise: an analysis at the European scale. Sci. Rep. 10, 11895. https://doi.org/10.1038/s41598-020-68576-0

Line 271: How are the forcing conditions defined? Which is the time interval? Do you apply a peak over threshold to identify extremes? If yes, why don't you use the event duration as one of the offshore parameters, but rather assume the same duration for all events? I would expect that events with larger wave heights will have larger duration thus more flooding.

Figure 4: This an important figure and I think that some improvements are needed:

- First of all, while you mention in the text that the density difference is due to the threshold used, I don't see why this should be the case. I would expect the lower values of the pairs to

have higher densities, since lower wave conditions would be more frequent. It would generally help to plot histograms for each variable at the x and y axis.

- The grey dots are the simulated samples. Why don't you plot the observed data as well? This would be critical to see if the simulated samples follow the structure of the observed data.
- The yellow dots (training data) should be plotted on top of the simulated points, since now sometimes they are not visible. Additionally, the training data do not seem to sample well the parameter space. This should not be the case if the maximum dissimilarity algorithm (MDA) was used.
- Some correlation statistics (Pearson or tail dependency) would really provide some insights on the dependencies between the parameters, which is one of the main points of the paper. Along with point 2, it would be nice to compare these indicators between observed and simulated pairs.
- In the caption you refer to the next section. This type of referring to parts of the manuscript that come afterwards happens in various points. Consider positioning the figures at a point were all things presented in the figure have been discussed already.

Lines 294-296 and Line 299: The way you select the extra cases with high SWL is not clear. How are these cases defined? Are they based on simulated conditions using an offset for SWL? This should be clarified in the text.

Table 1: Here, you present the performance indicators for the meta-models you present in Section 5, so it is not clear what they are about yet. Additionally, while you use a 10-fold validation I see only one value? Is this the average of the performance indicators? Shouldn't the range be included as well?

Figure 5: See my previous comment. Do you present all validations in the 10-fold validation here? Sometimes $Q^2$ is presented in decimal and other on percentage, try to be consistent. It would be interesting to see some validation with the actual flood volume instead of the logarithm as well.

Figure 6: What does this figure show exactly? Is this for the median SLR projections of RCP4.5? Which stochastics procedures are included here? What do the uncertainty bands describe exactly? Is this the total uncertainty of the projections of flood risk? Then one could question why the decomposition of the uncertainty is important if the uncertainty itself is that small.

Figure 11: For some of the light blue bars the widths used are different than that of the white bars ($H_s$, $T_p$, $D_p$). I imagine this can change the count that is plotted?

Line 498: "By 2100, the threshold…", from the graph the contribution of RCP seems minimal, while the DEM one is even larger than in 2050, so I am not sure why you mention the RCPs here.

**Technical corrections:**

Line 33: "…, flood severity is…"

Line 63: "… and to probabilistic assessments…", to is not needed here

Figure 1: Caption needs to be rephrased. Consider having a general title and then describing the panels. Moreover, there are things in the figure that are not described like the star and point P.

Line 120: Here, there is a reference to Fig.3 which has a caption where the next section is referred (Section 3.1). It would make more sense to show the figure when everything about it has already been described.

Line 123: Consider clarifying in the figure caption that the 50 random series are a subset of a larger number or realization that have been used to get the actual confidence bounds.

Line 197: If I am not mistaken $Q^2$ is commonly referred to as skill score. Maybe use that phrasing?

Line 202: "Let use first focus on the presentation by considering Y to…", what do the authors mean here?

Line 241: "In our study, the Shapley effect cannot be directly applied because the variable of interest Y is here not a scalar, but is binary and related to the flooding probability as defined in Eq.1". I thought that $Y$ is actually a scalar representing the water volume. Maybe rephrase?

Line 278: Here you express $u$ as a probability while in eq. 2 it is a continues value if I am not mistaken.

Line 285: "conditions (if they were independent any structure would have been noticed), hence" the parenthesis does not make sense. Maybe you actually want to say the opposite?

Line 324-326: Consider mentioning already for which $Y_c$ these values refer to.

Line 335: Here the authors jump to Fig.9 while the last figure was Fig.6.

Line 483: "The minor (with $Y_C$= 50m3), respectively very large (with $Y_C$=15,000m$^3$),…" this is not clear.

Figure 14: Why here there are no upper and lower bounds like in the other figures?

---

## Author Comment (AC1)

**Replies to Reviewer #1's comments on "Partitioning the uncertainty contributions of dependent offshore forcing conditions in the probabilistic assessment of future coastal flooding at a macrotidal site ". (nhess-2021-271)**

We would like to thank Reviewer #1 for the constructive comments. We agree with most of the suggestions and, therefore, we have modified the manuscript to take on board their comments. We recall here the reviews and we reply to each of the comments in turn.

**Reviewer #1:**

**General comment**

*The authors present an interesting methodological framework to assess the uncertainty contributions of forcing conditions to the probability of future flooding. Then, the framework is applied to a macrotidal study site at the French Atlantic coast, and its sensitivity to different framework assumptions is also tested.*
*The paper is quite interesting both in terms of proposed methods and yielded results, and will be of interest for NHESS readers.*
We thank Reviewer #1 for his/her positive analysis.

*The methodological framework is complex, and because of the way the paper is structured, some parts of the method are under the Section 4 (Application) instead of Section 3 (Methods). As a general comment, I would recommend a slight review of the current structure of the manuscript, trying to wrap all methodological information in section 3, leaving sections 4 and 5 to purely present results. A methodological chart might also help in such a complex study.*
Thank you for this suggestion. We have rewritten the methodological section in this sense (by moving any methodological description of Sect. 4 in Sect. 3). In this view, we have also rewritten the data description in Sect 2. Finally, Sect. 3.1 has been rewritten to better describe the different steps as well as the links between the sections, and a flowchart (new Figure 5) has been added to clarify the different steps.

[Figure]

**New Figure 5: Flowchart of the procedure. The sections describing the methods/data are indicated in grey next to the boxes.**

*In line with the previous comment, current section 6 presents a summary of results and future works. I would also recommend splitting this in Discussion and Conclusions. The results summary is repetitive and the manuscript could be improved with a broader discussion comparing the used methods (e.g. the Heffernan and Tawn 2004 approach, or the GPs) with other methods available in literature such as hierarchical copulas or RBFs (e.g. in Goulby et al., 2014). Some discussion on limitations is done in current chapter 6 although it seems short for such a complex study such as the one performed here, with many methodological steps.*

We agree with this suggestion. Sect. 5 is now dedicated to the discussion by assessing the impact of the modelling choices in Sect. 5.1 (as previously done) and the remaining limitations in Sect. 5.2 (regarding the modelling assumptions, the drivers of the flood processes and the *SLR* effect). In particular, we have shortened Sect. 5.1 by focusing on the description of the results of new Figure 9 (old Fig. 14) and by placing the details in Supplementary Materials E.

Regarding the use of alternative methods for extreme modelling, we have added this aspect in Sect. 5.2 by highlighting the interest of comparing to copula-based approaches; in particular by referring to the recent comparison exercise of Jane et al. (2020).

Regarding the use of alternative metamodelling techniques, we acknowledge that other methods could have been used. Though of interest, given the high predictive capability of the fitted GP (Q² >99%, see new Figure 6) in our case, we believe that this comparison would bring little added value. We preferably focus on the uncertainty related to the approximation of the true numerical model by a metamodel, i.e. the GP error. Contrary to other methods, GP can easily account for this type of error using the sampling-based approach described in Sect. 3.5. This is now better emphasized in Sect. 3.1. We have also underlined this aspect in the concluding remarks as well as in the abstract.

**Added reference:**
Jane, R.; Cadavid, L.; Obeysekera, J.; and Wahl, T.: Multivariate statistical modelling of the drivers of compound flood events in south Florida. Natural Hazards and Earth System Sciences, 20(10), 2681-2699, 2020.

**Specific comments**

*Line 33: "...flood severity is significantly increased..*
This is now corrected.

*Line 112: Maybe the sentence is lacking the verb: "..were built on…".*
This is now corrected.

*Line 266-270: This is general methodology. Consider defining the base case scenario and the alternative scenarios to test sensitivity on different assumptions under the methodological sections.*
Thank you for this suggestion. Given the comments of the two other reviewers, we preferably keep the original structure as such, because we believe that the readability is improved when the test sensitivity on different assumptions (originally described in Sect. 4) is discussed directly in the new Sect. 5 "discussion".

*Line 304: What does aggregating mean in this context? From what I understand, results of Q2 in Table 1 should be given as a mean +- std as they are the result of a 10-fold cross validation procedure. How is the single value given in table 1 computed from the 10 folds?*
We estimate a global performance indicator (here defined as the coefficient of determination Q²), and to do so we use the prediction errors calculated at all iterations of the cross-validation procedure. That is why there is only one value. Please refer to Hastie et al. (2009): Sect. 7.10 for further details. This is now clarified in Sect. 3 as follows.

"To validate the assumption of replacing the true numerical simulator by the kriging mean (Eq. 2a), we measure whether the GP model is capable of predicting "yet-unseen" input configurations, i.e. samples that have not been used for training. This can be examined by using a K-fold cross-validation approach (e.g. Hastie et al., 2009: Sect. 7.10). To do so, the training data is first randomly split into K roughly equal-sized parts. For the $k^{th}$ part, we fit the GP model to the other K−1 parts of the data, and calculate the prediction error of the fitted model when predicting the $k^{th}$ part of the data. We do this for k = 1,2,...,K and combine the K estimates of prediction error as follows.

Let us consider $\Lambda:\{1,...,n\} \rightarrow \{1,...,K\}$ an indexing function that indicates the partition's index to which each data point (of the training dataset) is allocated by the randomization, and denote by $\hat{m}_Y^{-k}(x)$ the prediction at $x$ using the GP model fitted using the $k^{th}$ part of the data removed. Then, the cross-validation estimate of the coefficient of determination denoted $Q^2$ holds as follows:

$$Q^2 = 1 - \frac{\sum_{i=1}^{i=n}\left(m_Y^i - \hat{m}_Y^{-\Lambda(i)}(x_i)\right)^2}{\sum_{i=1}^{i=n}(m_Y^i - \bar{m})^2}, \tag{3}$$

where $m_Y^i$ is the $i^{th}$ median value of *Y* computed using the modelling procedure of Sect. 2, and $\bar{m}$ is the average value of the numerically computed median values. A coefficient $Q^2$ close to 1.0 indicates that the GP model is successful in matching the new observations that have not been used for the training".

*Figure 5: is it showing 1 fold or the 10 folds? A word on the effect that the poorer performance of the GP approach around Yc = 50 m3 might have on the results might be interesting in the discussion.*

We confirm that new Figure 6 (old Figure 5) is showing 10 folds. We thank Reviewer #1 for noticing a possible problem around Yc = 50 m3. To check for any problem in our procedure, we have repeated the 10-fold cross validation procedure. New results are shown in new Figure 6: the same behavior can be noticed (though with some differences because the split of the dataset is done randomly as afore-described). Both figures clearly show a possible lack of predictability around Yc = 50 m3, and we now have clearly indicated in Sect. 4.1 that this potential problem is a motivation for accounting for the uncertainty in our results thanks to the procedure described in Sect. 3.5. The width of the error-bars in the Shapley effects' estimations confirm that the impact of this GP error is here only minor.

[Figure]

(a) Previous cross validation          (b) New cross validation

*Line 332-333: the ZCA-cor procedure and number of neighbors are both parameters of the R function, namely n.knn and rescale. This part and associated reference can be moved to the methods section, as it is confusing here.*
Thank you for this suggestion. This description is now placed in Sect. 3.5.

*Line 335. It is not standard to present Figure 9 before Figures 7 and 8. Consider moving this part later in the manuscript.*
We agree with this comment. The initial objective was to highlight the low uncertainty in the estimates of the Shapley effects, but this is only visible in Figure 9. To avoid referring to this figure, we propose to provide a new Table (Table 1 in Sect. 4.3) with the error estimates for the base case.

**Table 1. Shapley effects relative to the occurrence of the event $\{m_Y > Y_C = 2,000\text{m}^3\}$ given RCP4.5 scenario, estimated by computing the median value from B=50 replicates of the estimation procedure (Sect. 3.5) accounting for GP and sampling error. The numbers in brackets correspond to the minimum and maximum value computed from the estimation procedure.**

| Year | SLR | SWL | Hs | Tp | Dp | U | Du |
|------|-----|-----|-----|-----|-----|-----|-----|
| 2050 | 0.143 [0.126, 0.160] | 0.319 [0.293, 0.344] | 0.110 [0.095, 0.125] | 0.102 [0.086, 0.119] | 0.104 [0.095, 0.115] | 0.119 [0.104,0.130] | 0.104 [0.091, 0.112] |
| 2100 | 0.410 [0.398, 0.420] | 0.279 [0.270,0.291] | 0.082 [0.078, 0.085] | 0.045 [0.043, 0.047] | 0.047 [0.044, 0.051] | 0.091 [0.086, 0.096] | 0.047 [0.044, 0.050] |
| 2200 | 0.542 [0.536, 0.550] | 0.282 [0.277, 0.286] | 0.050 [0.048, 0.053] | 0.020 [0.019, 0.022] | 0.025 [0.022, 0.027] | 0.056 [0.053, 0.059] | 0.024 [0.022, 0.025] |

*Line 360-366: As in Line 266-270, this is general methodology. Consider defining the base case scenario and the alternative scenarios to test sensitivity on different assumptions under the methodological sections.*

Thank you for this suggestion. Given the comments of the two other reviewers, we preferably keep the original structure as such, because we believe that the readability is improved when the test sensitivity on different assumptions (originally described in Sect. 4) is discussed directly in the new Sect. 5 "discussion".

*Line 428. Do you men Figure 8a,b?*

Thank you for noticing this problem. This is now corrected.

Orleans,
March 28th, 2022
J. Rohmer[1] on behalf of the co-authors

[1] BRGM, 3 av. C. Guillemin - 45060 Orléans Cedex 2 – France

---

## Author Comment (AC2)

**Replies to Reviewer #2's comments on "Partitioning the uncertainty contributions of dependent offshore forcing conditions in the probabilistic assessment of future coastal flooding at a macrotidal site ". (nhess-2021-271)**

We would like to thank Reviewer #2 for the constructive comments. We agree with most of the suggestions and, therefore, we have modified the manuscript to take on board their comments. We recall the reviews and we reply to each of the comments in turn.

**Reviewer #2:**

*The paper "Partitioning the uncertainty contributions of dependent offshore forcing conditions in the probabilistic assessment of future coastal flooding at a macrotidal site" by Rohmer et al. presents an analysis of future flooding probability at Gâvres (France).*
*Lot of work have been put in the paper, which introduces several results that are exhaustively explained and discussed (perhaps even too much! see below). The research topic matches well the scope of the journal and the analysis presented seem to be correct.*
We thank Reviewer #2 for his/her positive analysis.

*However, there some flaws that make the manuscript hard to follow and some aspect must be clarified. Therefore, I believe that the paper should undergo a major revision before it is accepted for publication. See below for a list of comments:*
We agree with this comment. Below Reviewer #2 will find the description of the different modifications that we made; in particular to clarify the structure as well as our assumptions.

*The abstract is too long. If I remember correctly, NHESS recommends that it does not exceed 200 words. Perhaps you could shorten the paragraph starting at line 16; in fact, there is no need to provide so many details on the SLR contribution, given that you also introduce other variables further on the paper. Analogously, you could shorten the title (which is also rather long), removing information that are given in the text, e.g., the terms "uncertainty", "dependent", "at a macrotidal site" could be left off (but this is just a suggestion).*
Thank you for this suggestion. We have now shortened the abstract (total number of words = 200) as follows:

"Getting a deep insight into the role of coastal flooding drivers is of high interest for the planning of adaptation strategies for future climate conditions. Using global sensitivity analysis, we aim to measure the contributions of the offshore forcing conditions (wave/wind characteristics, still water level and sea level rise (*SLR*) projected up to 2200) to the occurrence of a flooding event at Gâvres town on the French Atlantic coast in a macrotidal environment. This procedure faces, however, two major difficulties, namely (1) the high computational time costs of the hydrodynamic numerical simulations; (2) the statistical dependence between the forcing conditions. By applying a Monte-Carlo-based approach combined with multivariate extreme value analysis, our study proposes a procedure to overcome these two difficulties by calculating sensitivity measures dedicated to dependent input variables (named Shapley effects) using Gaussian process (GP) metamodels. On this basis, our results show the increasing influence of *SLR* over time, and a small to moderate contribution of wave/wind characteristics,

or even negligible importance in the very long term. These results were discussed in relation to our modelling choices, in particular the climate change scenario, as well as the uncertainties of the estimation procedure (Monte Carlo sampling and GP error)."

We also propose a new title: "Partitioning the contributions of dependent offshore forcing conditions in the probabilistic assessment of future coastal flooding".

We choose to keep the term "dependent" because we believe that it is one major novelty of our work and we would like to emphasize this aspect.

*Why did you not account for river discharge? Isn't it relevant? You should at least provide some information on the mean discharge and explain why it is not considered in the analysis.* Thank you for this suggestion. The study is focused on the drivers of meteo-oceanic origin that participate to the occurrence of marine flooding. Yet, we acknowledge that other drivers exist and could also play a key role in the compound flooding like river discharge (in particular with the proximity of the Blavet river[1] on the study area) or rainfall. This is now underlined in Sect. 5.2 dedicated to discussing the limitations of our work as follows:

"Regarding the physical drivers of flooding, the analysis was focused on marine flooding by considering the joint effects of wave-wind-sea level, but additional processes are also expected to play a role in driving the compound flooding, like river discharge (in particular with the proximity of the Blavet river[2] on the study area) or rainfall. Including additional drivers is made here feasible by the flexibility of Heffernan and Tawn (2004)'s approach for analysing high dimensional extremes. This was shown in particular by Jane et al. (2020), who also highlighted the value of copula-based approaches, such as Vine copula. An avenue for future research could include the comparison of different approaches for multivariate extreme value analysis, i.e. a type of modelling uncertainty on top of the uncertainties in the parametrization and in the threshold selection of these techniques (e.g. Northrop et al., 2017)".

**Added references:**
Jane, R.; Cadavid, L.; Obeysekera, J.; and Wahl, T.: Multivariate statistical modelling of the drivers of compound flood events in south Florida. Natural Hazards and Earth System Sciences, 20(10), 2681-2699, 2020.
Northrop, P. J., Attalides, N., & Jonathan, P.: Cross-validatory extreme value threshold selection and uncertainty with application to ocean storm severity. Journal of the Royal Statistical Society: Series C (Applied Statistics), 66(1), 93-120, 2017.

*I believe that Sect. 2 and 3 should be reorganized, as right now it is hard to understand what has been done. The data should be introduced in a dedicated section, introducing the study area and describing the hindcast, current sea level, projected sea levels.*
*Next, the workflow should be detailed in a separated "Method" section, presenting first the hydrodynamic model and its validation (at least introduce appropriate references); next, the GP Metamodel should be described along with the selection of the events used to validate it against the hydrodynamic model. Then you can introduce the steps needed to force the validated GP Metamodel and analyze the results, that is Sect. 3.2 and 3.4. Finally, you could wrap up the section with a summary merged with what is now Sect. 3.5. Please make sure that all the*
* * *
[1] See Blavet gauge measurements (in French), https://www.vigicrues.gouv.fr/niv3-station.php?CdEntVigiCru=8&CdStationHydro=J571211004&GrdSerie=H&ZoomInitial=3
[2] See Blavet gauge measurements (in French), https://www.vigicrues.gouv.fr/niv3-station.php?CdEntVigiCru=8&CdStationHydro=J571211004&GrdSerie=H&ZoomInitial=3

*different bootstraps are clearly explained in the Methods section, i.e., the N realizations of the forcing conditions, the B repetitions to compute the Shapley effects, the 20 repetitions to mimic the variability of waves. In the current form of the paper, it is quite hard to understand the methodology.*

Thank you for this suggestion which is in agreement with the comments of the two other reviewers. We have rewritten the methodological section in this sense (by moving any methodological description of Sect. 4 in Sect. 3). In this view, we have also rewritten the data description in Sect 2. Sect. 3.1 has been rewritten to better describe the different steps as well as the links between the sections. Finally, a flowchart (new Figure 5) has been added to clarify the different steps.

[Figure]

**New Figure 5: Flowchart of the procedure. The sections describing the methods/data are indicated in grey next to the boxes.**

*Please fix the legend in Fig. 3. Lines of the 90% CI should be dashed if I understand correctly. Also, in panel a) it seems that the upper bound lies outside of the realization ensemble (it cannot be).*

Thank you for noticing this problem. It has been corrected. From Reviewer #2's comment, we understand that there is some confusion in the interpretation of Fig. 4 (old Fig. 3). The percentiles depicted in color in Fig. 4 are actually not calculated from the random samples but they are directly provided by Kopp et al. (2014): this is now better highlighted in the caption and this is also specified in the main text of Sect. 2. Furthermore to avoid any additional confusion, we have increased the number of random samples to better cover the space.

[Figure]

**New Figure 4: Future projection of regional *SLR* for 3 different RCP scenarios. The red line indicates the median, and the blue lines indicate the bounds of the 90% confidence interval provided by Kopp et al. (2014). The different black lines correspond to a subset of 75 randomly generated time series using the procedure described in Sect. 3.1.**

*Are you sure the offshore variables are dependent? In other words, is there the need to use a different equation rather than Eq. 5? Looking at the scatter plots of Fig. 4 I cannot appreciate any significant correlation pattern. Please provide some measure of the dependence between the time-series of different forcing.*

We agree with this comment. Indications of the evidence of the strong dependence have been added in Sect. 2.2 when presenting the offshore forcing conditions and in Sect. 4.2 when presenting the randomly generated samples. Full details of this dependence analysis have been provided in Supplementary Materials A. See below.

First, we provide the matrix of pairwise correlation coefficients (considering two types of correlation, i.e. Pearson and Kendall) calculated for the database of hindcasts (Sect. 2.2) used to perform the multivariate extreme value analysis. This clearly shows some dependencies.

[Figure]

**Figure A1. Pairwise correlation coefficients for the offshore forcing conditions (Pearson – left, Kendall - right)**

Second, we provide insights in the extremal dependence, i.e. the dependence when the considered variables take large values. To do so, we focus on the empirical estimates of $\bar{\chi}$ of summary statistics as defined by Coles et al. (1999) defined as follows:

$$\bar{\chi} = \lim_{u \to 1} \left( \frac{2\log(P(U>u))}{\log(P(U>u \cap V>u))} - 1 \right) \tag{A1}$$

where *U, V* are the two different forcing conditions, *u* is the quantile level. This indicator is used to screen locations where extremal dependence between both variables is exhibited: this is indicated where the $\bar{\chi}$ tends to 1.0 for very large quantile level *u* of both variables. The evaluation of $\bar{\chi}$ on the hindcast database shows that this remains below 1.0 in the limiting case hence indicating asymptotic independence. In this class of extremal dependence, $\bar{\chi}$ further provides a measure of the strength of dependence. Table A1 shows that this strength reaches

non negligible values (>0). The same analysis was conducted using the randomly generated samples (Table A2) and shows that the extremal dependence is well reproduced. We note that the differences are larger for ($H$s,$U$) but these can be considered satisfactory given the relatively large width of the confidence intervals (values in brackets).

**Table A1.** $\bar{\chi}$ **value for the hindcast database. Values in brackets correspond to the bounds of the 95% confidence interval**

|     | SWL | Hs |
|-----|-----|-----|
| Hs  | 0.28 (0.06, 0.50) |  |
| U   | 0.28 (0.06, 0.50) | 0.46 (0.20, 0.70) |

**Table A2.** $\bar{\chi}$ **value for the random generated samples. Values in brackets correspond to the bounds of the 95% confidence interval**

|     | SWL | Hs |
|-----|-----|-----|
| Hs  | 0.35 (0.10, 0.60) |  |
| U   | 0.25 (0.03, 0.48) | 0.70 (0.40, 1.00) |

Finally, we analyze in Table A3 in more details the values of the (a,b)-parameters (as defined in Eq. 2) of the dependence model. Note that the values should be read column-wise, which means that the value of the first column corresponds to the (a,b)- parameters when *SWL* is used as the conditional variable in Eq. 2. As discussed by Heffernan and Tawn (2004), their semi-parametric model allows to cover different types of extremal dependence with the following general rules:

For $2 \leq j \leq d$,
- When $a_j = 1$ and $b_j = 0$, the variables $(X_1, X_j)$ are asymptotically dependent;
- When $a_j < 1$, the variables $(X_1, X_j)$ are asymptotically independent.

In this latter case,
- When $0 < a_j < 1$ or $a_j = 0$ and and $b_j > 0$, means positive dependence;
- When $a_j = b_j = 0$, means near independence.

Table A3 clearly shows a non-negligible positive strength of dependence in the class of asymptotic independence (as indicated by $0 < a_j < 1$ and and $b_j > 0$).

**Table A3. (a,b)-parameters of the dependence model.**

|     | SWL | Hs | U |
|-----|-----|-----|-----|
| SWL |  | (0.348, 0.296) | (0.338, 0.352) |
| Hs  | (0.413, 0.327) |  | (0.662, 0.481) |
| U   | (0.167, 0.337) | (0.738, 0.359) |  |

**Added reference**
Coles, S., J. Heffernan, J. Tawn: Dependence Measures for Extreme Value Analyses. Extremes 2:4, 339–365, 1999.

*It is unclear to me why and how SWL and SLR contributions are split. Please clarify how you group the contributions of water levels between the two effects in the Methods section.*
This is now clarified in Sect. 2 as follows: "The analysis is conducted for future climate conditions by considering future still water level as $SWL_f(t) = SLR^{RCP}(t) + SWL$ where $SLR^{RCP}(t)$ is the value of sea level change in the future (relative to a given reference date) for

a given RCP scenario, and *SWL* is the present day still water level expressed with respect to the mean sea level of the considered reference date".

*Fourteen plots are a lot, make sure the total size does not exceed the journal recommendations while guaranteeing appropriate resolution of the figures. For example you could put Figures 9, 10, 12, 14 in the Supplement and summarize the related results in a single section. This way results would be more directly interpreted and their importance would be better framed.*
We agree with this comment. We have placed Figs.9,10,12,13 in Supplementary Materials E and have rewritten Sect 5.1 by focusing on the summary plot, new Fig. 9 (old Fig. 14).

*Please review the English grammar. I am not a native speaker but I found quite a few typos here and there.*
We thank Reviewer #2 for his/her careful reading. We have now double checked the grammar.

Orleans,
March 28th, 2022
J. Rohmer[1] on behalf of the co-authors

[1] BRGM, 3 av. C. Guillemin - 45060 Orléans Cedex 2 – France

---

## Author Comment (AC3)

**Replies to Reviewer #3's comments on "Partitioning the uncertainty contributions of dependent offshore forcing conditions in the probabilistic assessment of future coastal flooding at a macrotidal site ". (nhess-2021-271)**

We would like to thank Reviewer #3 for the constructive comments. We agree with most of the suggestions and, therefore, we have modified the manuscript to take on board their comments. We recall the reviews and we reply to each of the comments in turn.

**Reviewer #3:**

**General comments:**
*The manuscript "Partitioning the uncertainty contributions of dependent offshore forcing conditions in the probabilistic assessment of future coastal flooding at a macrotidal site" by Rohmer et al. presents an interesting framework to assess uncertainties in future flood risk projections at a site in France, which undoubtedly falls under the scope of NHESS.*
We thank Reviewer #3 for his/her positive analysis.

*While it is clear that a lot of work has gone in this study and manuscript preparation, there are some improvements that are needed before the manuscript should be accepted for publication. These should include some clarifications with respect to the methodology, an update in the structure of the paper and some language improvements. The methodology and results are quite interesting, but due to the complexity of the paper, the current structure is not optimal. For example, Section 4 named "Application" has a mixture of methods and results. Subsections 4.1 and 4.2 could be incorporated in section 3. Sections 2 and 3 could be merged and reorganized as well, starting with a presentation of the study site and available data, then the multivariate analysis, then describing the hydrodynamic model, then the GP meta-model and so on. A nice diagram with the methodological framework and references to the respective sections could really help the reader. This could be placed at the end of the introduction sections or in the beginning of the methodological section. Moreover, the figures are a lot and some of them (Figures 9-13) could go in the Appendix without any problem with the flow of the paper.*
Thank you for this suggestion which is in agreement with the comments of the two other reviewers. We have rewritten the methodological section in this sense (by moving any methodological description of Sect. 4 in Sect. 3). In this view, we have also rewritten the data description in Sect 2. Sect. 3.1 has been rewritten to better describe the different steps as well as the links between the sections. Finally, a flowchart (new Figure 5) has been added to clarify the different steps. Finally, we have placed Figs.9,10,12,13 in Supplementary Materials E and have rewritten Sect 5.1 by focusing on the description of the summary plot, new Fig. 9 (old Fig. 14).

[Figure]

**New Figure 5: Flowchart of the procedure. The sections describing the methods/data are indicated in grey next to the boxes.**

*While the dependency of the offshore parameters seems to be one of the main highlights of the paper, the presentation of this part and especially Figure 4, are not that clear. Some extra dependency indicators and improvements in the figure could help to clarify this (see my comments bellow for more specific information).*

We agree with this comment. Indications of the evidence of the strong dependence have been added in Sect. 2.2 when presenting the offshore forcing conditions and in Sect. 4.2 when presenting the randomly generated samples. Full details of this dependence analysis have been provided in Supplementary Materials A. See below.

First, we provide the matrix of pairwise correlation coefficients (considering two types of correlation, i.e. Pearson and Kendall) calculated for the database of hindcasts (Sect. 2.2) used to perform the multivariate extreme value analysis. This clearly shows some dependencies.

[Figure]

**Figure A1. Pairwise correlation coefficients for the offshore forcing conditions (Pearson – left, Kendall - right)**

Second, we provide insights in the extremal dependence, i.e. the dependence when the considered variables take large value. To do so, we focus on the empirical estimates of $\bar{\chi}$ of summary statistics as defined by Coles et al. (1999) defined as follows:

$$\bar{\chi} = \lim_{u \to 1} \left( \frac{2\log(P(U>u))}{\log(P(U>u \cap V>u))} - 1 \right) \tag{A1}$$

where $U, V$ are the two different forcing conditions, $u$ is the quantile level. This indicator is used to screen locations where extremal dependence between both variables is exhibited: this is indicated where the $\bar{\chi}$ tends to 1.0 for very large quantile level $u$ of both variables. The evaluation of $\bar{\chi}$ on the hindcast database shows that this remains below 1.0 in the limiting case hence indicating asymptotic independence. In this class of extremal dependence, $\bar{\chi}$ further provides a measure of the strength of dependence. Table A1 shows that this strength reaches non negligible values ($>0$). The same analysis was conducted on the randomly generated samples (Table A2) and shows that the extremal dependence is well reproduced. We note that the differences are larger for ($Hs,U$) but these can be considered satisfactory given the relatively large width of the confidence intervals (values in brackets).

**Table A1. $\bar{\chi}$ value for the database of hindcast databases. Values in brackets correspond to the bounds of the 95% confidence interval**

|  | *SWL* | *Hs* |
|---|---|---|
| Hs | 0.28 (0.06, 0.50) | |
| U | 0.28 (0.06, 0.50) | 0.46 (0.20, 0.70) |

**Table A2. $\bar{\chi}$ value for the random generated samples. Values in brackets correspond to the bounds of the 95% confidence interval**

|  | SWL | *H*s |
|---|---|---|
| *H*s | 0.35 (0.10, 0.60) | |
| *U* | 0.25 (0.03, 0.48) | 0.70 (0.40, 1.00) |

Finally, we analyze in more details in Table A3 the values of the (a,b)-parameters (as defined in Eq. 2) of the dependence model. Note that the values should be read columnwise, which means that the value of the first column correspond to the (a,b)- parameters when *SWL* is used as the conditional variable in Eq. 2. As discussed by Heffernan and Tawn (2004), their semi-parametric model allows to cover different types of extremal dependence with the following general rules:

For $2 \leq j \leq d$,
- When $a_j=1$ and $b_j=0$, the variables $(X_1, X_j)$ are asymptotically dependent;
- When $a_j<1$, the variables $(X_1, X_j)$ are asymptotically independent.
In this latter case,
- When $0<a_j<1$ or $a_j=0$ and and $b_j>0$, means positive dependence;
- When $a_j=b_j=0$, means near independence.

Table A3 clearly shows a non-negligible positive strength of dependence in the class of asymptotic independence (as indicated by $0<a_j<1$ and and $b_j>0$).

**Table A3. (a,b)-parameters of the dependence model.**

|  | *SWL* | Hs | U |
|---|---|---|---|
| *SWL* | | (0.348, 0.296) | (0.338, 0.352) |
| Hs | (0.413, 0.327) | | (0.662, 0.481) |
| U | (0.167, 0.337) | (0.738, 0.359) | |

**Added reference**

Coles, S., J. Heffernan, J. Tawn: Dependence Measures for Extreme Value Analyses. Extremes 2:4, 339–365, 1999.

*Moreover, I would expect that the duration of an offshore event is an important parameter when assessing flood risk. However, the authors use a fixed duration of 20 minutes with uniform conditions. I would appreciate if the authors justify this choice.*

In this study, we use numerical simulations with steady state offshore conditions as described in Sect. 2.1 as follows: "The inland impact of the storm event is assessed by estimating the total water volume $Y$ that has entered the territory at high tide. This is performed by first running the WW3 model (over 2 hours to reach steady wave conditions), and then the SWASH model by considering a time span of 20 minutes (with 5 minutes spin up) and steady state offshore forcing conditions. The value of $Y$ is the volume at the end of the simulation. Such simulation costs about 1h30 of time computation on 48 cores approximately."

We acknowledge that it is a limitation of our procedure and we have added a discussion in Sect. 5.2 as follows:

"Regarding the modelling of the flood processes, one of our main assumptions is to perform simulations with steady state offshore forcing conditions, i.e. without accounting for the time evolution of the forcing conditions around the high tide (Sect. 2.1). First, this choice was guided by the computational budget that could be afforded to account for wave stochasticity via repeated numerical simulations. A total of 144×20=2,880 numerical simulations were performed here for our analysis: such a large number of simulations would be difficult to achieve using non-stationary numerical simulations, because a single run takes about 3 days of computation on 48 cores. Second, Idier et al. (2020b) showed, on two historical storm events, that the value of $Y$ remains of the same order of magnitude between a steady-state and a non-stationary simulation. Therefore, the temporal effect is expected to influence only moderately our conclusions regarding uncertainty partitioning. If, however, other flooding indicators are chosen (e.g. total flooded area, or water height at a given inland location), i.e. indicators that are more sensitive to the time evolution of offshore conditions, non-stationary simulations are mandatory. In this case, time dimension should be accounted for at different levels of the procedure: (1) metamodelling with functional inputs (e.g. using the procedure developed by Betancourt et al., 2020); (2) integrating additional variables in the multivariate extreme value analysis like event duration and event spacing (e.g. Callaghan et al., 2008); (3) random generation of time-varying forcing conditions (e.g. using the stochastic emulator used by Cagigal et al. (2020) to force ensemble long-term shoreline predictions)".

**Added references**
Betancourt, J.; Bachoc, F.; Klein, T.; Idier, D.; Pedreros, R.; and Rohmer, J.: Gaussian process metamodeling of functional-input code for coastal flood hazard assessment. Reliability Engineering & System Safety, 198, 106870, 2020.
Cagigal, L.; Rueda, A.; Anderson, D.L.; Ruggiero, P.; Merrifield, M.A.; Montaño, J.; Coco, G.; Méndez, F.J.; A multivariate, stochastic, climate-based wave emulator for shoreline change modelling. Ocean Model. 154, 2020.
Callaghan, D. P., Nielsen, P., Short, A., and Ranasinghe, R.W.M.R.J.B.: Statistical simulation of wave climate and extreme beach erosion. Coastal Engineering, 55(5), 375-390, 2008.

*Additionally, I feel that for such a complex methodological framework the discussion (including limitations sections) is rather short. Maybe section 5 could actually go in the discussion section (I feel like it can belong there since it discusses the assumptions used), especially if the figures*

*that accompany it, are moved to the appendix. Other limitations and assumptions should be discussed as well; the way the dependency structure is modelled relatively to other methods available like e.g., copulas; the use of GP as a meta-model versus other statistical techniques etc.*

We agree with this suggestion. Sect. 5 is now dedicated to the discussion by assessing the impact of the modelling choices in Sect. 5.1 (as previously done) and the remaining limitations in Sect. 5.2 (regarding the modelling assumptions, the drivers of the flood processes and the *SLR* effect). In particular, we have shortened Sect. 5.1 by focusing on the description of the results of new Figure 9 (old Fig. 14) and by placing the details in Supplementary Materials E.

Regarding the use of alternative methods for extreme modelling, we have added this aspect in Sect. 5.2 by highlighting the interest of comparing to copula-based approaches; in particular by referring to the recent comparison exercise of Jane et al. (2020) as follows:

"Regarding the physical drivers of flooding, the analysis was focused on marine flooding by considering the joint effects of wave-wind-sea level, but additional processes are also expected to play a role in driving the compound flooding, like river discharge (in particular with the proximity of the Blavet river[1] on the study area) or rainfall. Including additional drivers is made here feasible by the flexibility of Heffernan and Tawn (2004)'s approach for analysing high dimensional extremes. This was shown in particular by Jane et al. (2020), who also highlighted the value of copula-based approaches, such as Vine copula. An avenue for future research could include the comparison of different approaches for multivariate extreme value analysis, i.e. a type of modelling uncertainty on top of the uncertainties in the parametrization and in the threshold selection of these techniques (e.g. Northrop et al., 2017)".

Regarding the use of alternative metamodelling techniques, we acknowledge that other methods could have been used. Though of interest, given the high predictive capability of the fitted GP ($Q^2$ >99%, see new Figure 6) in our case, we believe that this comparison would bring little added value. We preferably focus on the uncertainty related to the approximation of the true numerical model by a metamodel, i.e. the GP error. Contrary to other methods, GP can easily account for this type of error using the sampling-based approach described in Sect. 3.5. This is now indicated in Sect. 3.1. We have also underlined this aspect in the concluding remarks as well as in the abstract.

**Added reference:**
Jane, R.; Cadavid, L.; Obeysekera, J.; and Wahl, T.: Multivariate statistical modelling of the drivers of compound flood events in south Florida. Natural Hazards and Earth System Sciences, 20(10), 2681-2699, 2020.
Northrop, P. J., Attalides, N., & Jonathan, P.: Cross-validatory extreme value threshold selection and uncertainty with application to ocean storm severity. Journal of the Royal Statistical Society: Series C (Applied Statistics), 66(1), 93-120, 2017.

*In general, the language could improve as well, as I noticed there were some grammar issues and typos here and there.*
We thank Reviewer #3 for his/her careful reading. We have now double checked the grammar and typos.
* * *
[1] See Blavet gauge measurements (in French), https://www.vigicrues.gouv.fr/niv3-station.php?CdEntVigiCru=8&CdStationHydro=J571211004&GrdSerie=H&ZoomInitial=3

**Specific comments:**

*Line 54: Athanasiou et al. 2020 applied GSA as well for coastal erosion projections at the European Scale*

*Athanasiou, P., van Dongeren, A., Giardino, A., Vousdoukas, M.I., Ranasinghe, R., Kwadijk, J., 2020. Uncertainties in projections of sandy beach erosion due to sea level rise: an analysis at the European scale. Sci. Rep. 10, 11895. https://doi.org/10.1038/s41598-020-68576-0*

Thank you for the suggestion; this has been added to the reference list.

*Line 271: How are the forcing conditions defined? Which is the time interval? Do you apply a peak over threshold to identify extremes? If yes, why don't you use the event duration as one of the offshore parameters, but rather assume the same duration for all events? I would expect that events with larger wave heights will have larger duration thus more flooding.*

In Sect. 2.2, we now clarify how the conditions are selected as follows: "A total of >80,000 past events characterized by sixplets (*SWL*, *Hs*, *U*, *Tp*, *Dp*, and *Du*) taken at the time instant of the high tide, are used in the following to constrain the statistical methods of Sect. 3."

In our study, we use numerical simulations with steady state conditions i.e. without accounting for the temporal evolution around the high tide. See our reply above. We agree that accounting for the duration of the events is an interesting line for future research. This is now clearly indicated in the discussion section, Sect. 5.2.

*Figure 4: This an important figure and I think that some improvements are needed:*
Thank you for these suggestions. A new Figure 3 (old Figure 4) has now been added.

[Figure]

**New Figure 3: Overview of the *N*=50,000 randomly generated samples of offshore conditions (red dots). Black dots correspond to the hindcast conditions used to fit the statistical methods described in Sect. 3.3. Blue (open and filled) circles correspond to the *n*=144 training data used to set-up the GP metamodels. The open circles correspond to cases that are deliberately selected outside the range of the red dots to cover a broad range of situations (the selection approach is detailed in Sect. 3.2).**

- *First of all, while you mention in the text that the density difference is due to the threshold used, I don't see why this should be the case. I would expect the lower values of the pairs to have higher densities, since lower wave conditions would be more frequent. It would generally help to plot histograms for each variable at the x and y axis.*

To fulfil this recommendation, we have added the marginals in Supplementary Materials B as well as the observations in Fig. 3 (black dots). However, we believe that adding the histograms would make Figure 3 less clear. Besides, it should be noted that the interest of the multivariate extreme value analysis is to extrapolate the values towards large values, hence the histograms would reflect this "correction", which would make the interpretation even less straightforward. Therefore, we chose not to add them.

- *The grey dots are the simulated samples. Why don't you plot the observed data as well? This would be critical to see if the simulated samples follow the structure of the observed data.*
  See new Figure 3.

- *The yellow dots (training data) should be plotted on top of the simulated points, since now sometimes they are not visible. Additionally, the training data do not seem to sample well the parameter space. This should not be the case if the maximum dissimilarity algorithm (MDA) was used.*
  See new Figure 3. Regarding the distribution of the samples, the construction has not been done by using MDA only but by combining two approaches: (1) for the extreme values, we apply the MDA-based approach developed by Gouldby et al. (2014); (2) for low and moderate values, we apply the conditioned latin hypercube sampling procedure of Minasny and McBratney (2006). Note that the selection procedure is beyond the scope of the present study and the reader can refer to Rohmer et al. (2020) for further details on the implementation for the site of interest here. This is also indicated.
  In addition, 44 extra samples were added based on a complementary study described by Idier et al. (2021). The randomly generated cases with large *SWL* values that lead to a positive water height at the observation point P (Fig. 1b) were selected (See below).

  Therefore, the samples do not necessarily cover the whole parameter space as would be the case with the direct application of MDA.

- *Some correlation statistics (Pearson or tail dependency) would really provide some insights on the dependencies between the parameters, which is one of the main points of the paper. Along with point 2, it would be nice to compare these indicators between observed and simulated pairs.*
  We agree with this comment. See our reply above.

- *In the caption you refer to the next section. This type of referring to parts of the manuscript that come afterwards happens in various points. Consider positioning the figures at a point were all things presented in the figure have been discussed already.*
  Thank you for noticing this problem which is now corrected.

*Lines 294-296 and Line 299: The way you select the extra cases with high SWL is not clear. How are these cases defined? Are they based on simulated conditions using an offset for SWL? This should be clarified in the text.*
The extra 44 additional cases (open blue circles in Fig. 3) were defined using the set of high tide conditions that were randomly generated for the design of the early-warning system at Gâvres (Idier et al. (2021): Sect. 2.5). These conditions were used as inputs of the metamodel implemented by Rohmer et al. (2020) to predict the flooding-induced water height at the observation point P (Fig. 1b); those leading to a positive water height were then selected.

The selection procedure of the 100 cases is also clarified in Sect. 3.2 as follows: "The *n* numerical experiments used to train the GP model are selected based on the combination of two approaches: (1) for the extreme values, we apply the approach developed by Gouldby et al. (2014) by applying clustering algorithms to a large data set of extreme forcing conditions. This database is constructed through a combination of Monte Carlo random sampling and multivariate extreme value analysis performed on the database of hindcast conditions described in Sect. 2.2; (2) for low and moderate values, we apply the conditioned latin hypercube sampling procedure of Minasny and McBratney (2006)."

Note that the selection procedure is beyond the scope of the present study and the reader can refer to Rohmer et al. (2020) for further details on the implementation for the site of interest here. This is also indicated.

**Added references**
Idier, D., Aurouet, A., Bachoc, F., Baills, A., Betancourt, J., Gamboa, F., et al.: A User-Oriented Local Coastal Flooding Early Warning System Using Metamodelling Techniques. Journal of Marine Science and Engineering, 9(11), 1191, 2021.
Minasny, B.; McBratney, A. B.: A conditioned Latin hypercube method for sampling in the presence of ancillary information. Computers & geosciences 32(9):1378-1388, 2006.

*Table 1: Here, you present the performance indicators for the meta-models you present in Section 5, so it is not clear what they are about yet. Additionally, while you use a 10-fold validation I see only one value? Is this the average of the performance indicators? Shouldn't the range be included as well?*
We estimate a global performance indicator (here defined as the coefficient of determination Q²), and to do so we use the prediction errors calculated at all iterations of the cross-validation procedure. Therefore, there is only one single value.

Please refer to Hastie et al. (2009): Sect. 7.10 (in particular equation 7.48) for further details. This is now clarified in Sect. 3 as follows.

"To validate the assumption of replacing the true numerical simulator by the kriging mean (Eq. 2a), we measure whether the GP model is capable of predicting "yet-unseen" input configurations, i.e. samples that have not been used for training. This can be examined by using a K-fold cross-validation approach (e.g. Hastie et al., 2009: Sect. 7.10). To do so, the training data is first randomly split into K roughly equal-sized parts. For the $k^{th}$ part, we fit the GP model to the other K−1 parts of the data, and calculate the prediction error of the fitted model when predicting the $k^{th}$ part of the data. We do this for k = 1,2,...,K and combine the K estimates of prediction error as follows.

Let us consider $\Lambda:\{1,\dots,n\} \to \{1,\dots,K\}$ an indexing function that indicates the partition's index to which each data point (of the training dataset) is allocated by the randomization, and denote by $\widehat{m}_Y^{-k}(x)$ the prediction at $x$ using the GP model fitted using the $k^{th}$ part of the data removed. Then, the cross-validation estimate of the coefficient of determination denoted $Q^2$ holds as follows:

$$Q^2 = 1 - \frac{\sum_{i=1}^{j=n}\left(m_Y^i - \widehat{m}_Y^{-\Lambda(i)}(x_i)\right)^2}{\sum_{i=1}^{j=n}\left(m_Y^i - \bar{m}\right)^2}, \tag{3}$$

where $m_Y^i$ is the $i^{th}$ median value of $Y$ computed using the modelling procedure of Sect. 2, and $\bar{m}$ is the average value of the numerically computed median values. A coefficient $Q^2$ close to 1.0 indicates that the GP model is successful in matching the new observations that have not been used for the training".

*Figure 5: See my previous comment. Do you present all validations in the 10-fold validation here? Sometimes Q2 is presented in decimal and other on percentage, try to be consistent. It would be interesting to see some validation with the actual flood volume instead of the logarithm as well.*

We confirm that we present 10 folds following the afore-described procedure. Reviewer #1 pointed out a possible problem with our procedure because of the deviation at $Yc=50m3$. To check for any problem in our procedure, we have repeated the 10-fold cross validation procedure a second time. New results are shown in new Figure 6 (old Fig. 5) and clearly indicate the same behavior (though some differences are noticeable because the split of the dataset is done randomly as afore-described). Both figures show a possible lack of predictability in this region. Thus, we have clearly indicated in Sect. 4.1 that this potential problem is a motivation for accounting for this uncertainty in our results thanks to the procedure described in Sect. 3.5. The width of the error-bars in the Shapley effects' estimations (see new Table 1) confirm that the impact of this GP error is here only minor.

[Figure]

(a) Previous cross validation         (b) New cross validation

Finally we justified in Sect. 4.1 the use of the log-transformation as follows: "Due to the highly skewed distribution of $m_Y$, we use a logarithm transformation i.e. $\log_{10}(m_Y + 1)$. Our preliminary tests also showed that this transformation improved the predictive capability of the metamodel by increasing $Q^2$ by 10%".

*Figure 6: What does this figure show exactly? Is this for the median SLR projections of RCP4.5? Which stochastics procedures are included here? What do the uncertainty bands describe exactly? Is this the total uncertainty of the projections of flood risk? Then one could question why the decomposition of the uncertainty is important if the uncertainty itself is that small.*

The caption of New Figure 7 (old Fig. 6) has been rewritten to improve the presentation as follows: "Time evolution of the flooding probability that the median value of the inland water volume induced by the flood exceeds the threshold of $Y_C$=2,000m$^3$ given *SLR* projections for the scenario RCP4.5. The inserted figure indicates the very small uncertainty band's width whose limits are the lower and upper bounds computed using $B$=50 replicates of the estimation procedure (Sect. 3.5) accounting for GP and sampling error".

It is important to note that the uncertainty band is not the one decomposed using the Shapley effects, but it is the uncertainty of the procedure, i.e. the GP and sampling error as described in Sect. 3.5. The Shapley effects do not decompose the uncertainty on the flooding probability but decompose the uncertainty on the indicator function that defines the flooding probability. This means that we measure the contribution of the forcing conditions to the occurrence of a flooding event. This is now clarified in Sect. 3.4 by first presenting the problem for the most commonly used case (i.e. decomposition of the variance of a continuous random variable) and then, the adaptation to tackle the problem of flooding event's occurrence. Note that throughout the text, we paid attention to provide this interpretation of the sensitivity analysis.

Having this interpretation in mind, a small uncertainty band's width is here good news because it indicates low uncertainty related to our procedure.

*Figure 11: For some of the light blue bars the widths used are different than that of the white bars (Hs, Tp, Dp). I imagine this can change the count that is plotted?*
This Figure is now placed in Supplementary Materials E. The initial objective was to highlight that the mode and dispersion were different between both cases: the change of the bins did not impact this conclusion.

*Line 498: "By 2100, the threshold…", from the graph the contribution of RCP seems minimal, while the DEM one is even larger than in 2050, so I am not sure why you mention the RCPs here.*
Thank you for pointing out this confusion. We have rephrased as follows: "In the long term (by 2100), the threshold importance becomes significant, and it is only in the very long term (by 2200) that the RCP scenario starts to play a key role; in particular in relation to the higher mode of the *SLR* probability distribution for RCP8.5 scenario."

**Technical corrections:**
*Line 33: "…, flood severity is…"*
This has been corrected.

*Line 63: "… and to probabilistic assessments…", to is not needed here*
This has been corrected.

*Figure 1: Caption needs to be rephrased. Consider having a general title and then describing the panels. Moreover, there are things in the figure that are not described like the star and point P.*
The star and point "P" have now been removed. The caption has been corrected as follows: "Digital Elevation Model (DEM) and computational domain of the study site of Gâvres for the spectral wave model WW3 (a), and for the non-hydrostatic phase-resolving model SWASH (b). The insert in (a) provides the regional setting. The point P indicates an observation point on the coast. Adapted from Idier et al. (2020a)".

*Line 120: Here, there is a reference to Fig.3 which has a caption where the next section is referred (Section 3.1). It would make more sense to show the figure when everything about it has already been described.*
We agree with this comment and we have now placed the description of the selection of *SLR* time series in Sect. 2.

*Line 123: Consider clarifying in the figure caption that the 50 random series are a subset of a larger number or realization that have been used to get the actual confidence bounds.*

As also pointed out by Reviewer #2, there is some confusion in the interpretation of new Fig 4 (old Fig. 3). The percentiles depicted in color in Fig. 4 are actually not calculated from the random samples but they are directly provided by Kopp et al. (2014): this is now better highlighted in the caption and this is also specified in the main text of Sect. 2. Furthermore to avoid some additional confusion we have increased the number random samples to better cover the space.

[Figure]

**New Figure 4: Future projection of regional *SLR* for 3 different RCP scenarios. The red line indicates the median, and the blue lines indicate the bound of the 90% confidence interval provided by Kopp et al. (2014). The different black lines correspond to a subset of 75 randomly generated time series using the procedure described in Sect. 3.1.**

*Line 197: If I am not mistaken Q2 is commonly referred to as skill score. Maybe use that phrasing?*

Thank you for this suggestion. In the computer experiment community, this indicator is preferably named coefficient of determination. This is now specified in Sect. 3.

*Line 202: "Let use first focus on the presentation by considering Y to…", what do the authors mean here?*

*Line 241: "In our study, the Shapley effect cannot be directly applied because the variable of interest Y is here not a scalar, but is binary and related to the flooding probability as defined in Eq.1". I thought that Y is actually a scalar representing the water volume. Maybe rephrase?*

Thank you for this comment which highlights that our description is not clear enough. Our initial objective was to first present the Shapley effects in the most-commonly used case i.e. for a continuous random variable and then to highlight the adaptation to handle the case of the exceedance event. To improve the presentation, we have now split Sect. 3.4 into two sub-sections and have added the following introductive paragraph.

"The objective is to investigate the influence of the offshore conditions with respect to the occurrence of the event $\{m_Y > Y_C\}$ in relation to the flooding probability defined in Eq. 1. To do so, we opt for the adaptation of the GSA approach based on the Shapley effects proposed by Idrissi et al. (2021) in the domain of reliability assessment. For sake of presentation clarity, we first present the Shapley effect by considering the classical situation where the variable of interest is a scalar variable (Sect. 3.4.1). Second, we present the adaptation in relation to the problem of flooding probability (Sect. 3.4.2)".

*Line 278: Here you express u as a probability while in eq. 2 it is a continues value if I am not mistaken.*

Here the threshold denoted $u$ is defined for defining the marginals and is expressed in terms of physical units. It is the threshold denoted $\upsilon$ of the dependence model (Eq. 2) that is expressed in probabilistic terms.

*Line 285: "conditions (if they were independent any structure would have been noticed), hence" the parenthesis does not make sense. Maybe you actually want to say the opposite?*
Thank you for noticing this problem. This has been corrected as follows: "The visual analysis of the extracted conditions (black dots in Fig. 3) suggests a moderate-to-large statistical dependence between the forcing conditions, because we can clearly see a structure between the points: if they were independent, no structure would be noticed."

*Line 324-326: Consider mentioning already for which Yc these values refer to.*
This is now underlined in Sect. 4.3.

*Line 335: Here the authors jump to Fig.9 while the last figure was Fig.6.*
Thank you for noticing this problem which has now been corrected.

*Line 483: "The minor (with YC= 50m3), respectively very large (with YC=15,000m3),…" this is not clear.*
This part has been removed.

*Figure 14: Why here there are no upper and lower bounds like in the other figures?*
We do not use any error bar, because here we aim to plot the relative differences of the Shapley effect for *SLR*. This is done by using the median value computed for $B$=50 replicates of the estimation procedure with respect to the base case defined in Sect. 4. This is now indicated in the caption of new Figure 9 (old Figure 14). The uncertainty on the different estimates of the Shapley effects are provided in the different figures now placed in Supplementary Materials E.

Orleans,
March 28[th], 2022
J. Rohmer[1] on behalf of the co-authors

[1] BRGM, 3 av. C. Guillemin - 45060 Orléans Cedex 2 – France

---

## Referee Report (RR1)

**General comments:**

I would like to thank the authors for revising the manuscript and taking into account my comments. Unquestionably, this is a quite improved version relative to the previous one. However, there are still some points that I would like to be clarified before accepting this manuscript for publication.

In general, I am happy with how the authors responded to my comments on the dependency structure and the duration of the events. With respect to the structure of the paper I think there is still room for improvement and Figure 5 could be a bit clearer.

The main point that is still not clear to me is the way the Monte-Carlo simulation is performed to obtain the events used for the flooding probability simulations. Bellow I have more specific comments on this. Additionally, I would still like to see a different way of calculating the error statistics of the meta-model.

**Specific comments:**

**Figure 3:**

The blue colour for the selected events might be a poor choice with respect to visibility, against the black colour of the observed events.

My main observation though is on the simulated events, which are still puzzling me. That is why I asked to see the histograms in my previous comment.

You mention:

Line 223: "Step (1) Fitting of the marginals of 'amplitude' variables through the combination of the empirical distribution, below a suitable high threshold $u$, and of the Generalised Pareto distribution (GPD) above the selected threshold $u$ (Coles and Tawn, 1991) using the method of moments."

Line 347: "Following Step (1) described in Sect. 3.3, the extracted data are used to fit the marginals of the 'amplitude' variables using the GPD distribution with the selected threshold value $u_{Hs}=6.2m$, $u_{Skew Surge}=0.48m$, and $u_U=18.9m/s$ corresponding to ~2 extreme events / year. The marginal distributions are provided in Supplementary Materials B."

Line 355: "Note that some delineations (on the bottom left hand corner) can be noticed, which results from the threshold-based procedure to model the probabilistic distributions (see Sect. 3.3)."

First of all, I would expect that the thresholds values you provided would match the delineations seen in the figure, which from what I can see in Figure 3, are around 4 m, 13 m/s and 2.3 m for Hs, U and SWL respectively.

The simulated events seem to be quite dense above these thresholds (i.e., extremes events), while on the opposite side there are not that many events bellow the threshold. In essence, you are simulating more extreme than mild events, which I would not expect to be the case. Please explain if I am missing something here. This would mean that your probabilistic analysis for the flood volume is affected as well.

I have mentioned this issue in my previous review as well, but I did not recieve any response from the authors on this.

**Figure 5:**

It is still a bit challenging to navigate this figure. For example the n training data used in STEP 1 are subsampled by the output of STEP 2. I would expect that a change of order in STEP 1 and 2 will make this clearer.

I think it will help to add some general titles in the steps as well, like for example "Meta-model for flood volume estimation", "Monte-Carlo sampling", and "Global sensitivity analysis" for the current steps 1-3.

It is not explained (and I do not understand) why some boxes have more intense colours.

**Figure 6:**

Thank you for clarifying why there is only a single value for $Q^2$ and not a mean with an uncertainty range. Yet, the method you are using to calculate that (Hastie et al., 2009, which I noticed is missing in the reference list in the end of the manuscript) is used in Hastie et al., 2009 to calculate an estimate of the prediction error (based on a loss function). While here you are applying this to calculate an error-statistic ($Q^2$) that is affected by the variance of the observed (modelled) data, which since you are merging all the k-folds together is quite high. Hence, I would propose to calculate $Q^2$ for each of the k-folds, and then present the average values with an uncertainty band. I am curious to see if the mean $Q^2$ will be as high as 99.2%.

**Minor comments:**

**Line 351:** "**...**threshold ʋ of Eq. (2)..."

I imagine this is equation 4 now

**Line 353**: "N=50,000 events (representative of 1,000 years)."

How do you scale the number of events to number of years?

---

## Author Response (AR2)

**Replies to Reviewer #3's comments on "Partitioning the uncertainty contributions of dependent offshore forcing conditions in the probabilistic assessment of future coastal flooding at a macrotidal site". (nhess-2021-271)**

**Reviewer #3:**

**General comments:**

I would like to thank the authors for revising the manuscript and taking into account my comments. Unquestionably, this is a quite improved version relative to the previous one. However, there are still some points that I would like to be clarified before accepting this manuscript for publication.

In general, I am happy with how the authors responded to my comments on the dependency structure and the duration of the events. With respect to the structure of the paper I think there is still room for improvement and Figure 5 could be a bit clearer.

The main point that is still not clear to me is the way the Monte-Carlo simulation is performed to obtain the events used for the flooding probability simulations. Bellow I have more specific comments on this. Additionally, I would still like to see a different way of calculating the error statistics of the meta-model.

We would like to thank Reviewer #3 for the valuable and constructive comments. We agree with the suggestions regarding the Monte-Carlo-based approach as well as regarding the validation of the meta-model. Therefore, we have modified the manuscript to take on board the comments. We recall here the reviews and we reply to each of the comments in turn.

**Specific comments:**

**Figure 3:**

The blue colour for the selected events might be a poor choice with respect to visibility, against the black colour of the observed events.

My main observation though is on the simulated events, which are still puzzling me. That is why I asked to see the histograms in my previous comment.

You mention:

- Line 223: "Step (1) Fitting of the marginals of 'amplitude' variables through the combination of the empirical distribution, below a suitable high threshold u, and of the Generalised Pareto distribution (GPD) above the selected threshold u (Coles and Tawn, 1991) using the method of moments."
- Line 347: "Following Step (1) described in Sect. 3.3, the extracted data are used to fit the marginals of the 'amplitude' variables using the GPD distribution with the selected threshold value uHs=6.2m, uSkew Surge=0.48m, and uU=18.9m/s corresponding to ~2 extreme events / year. The marginal distributions are provided in Supplementary Materials B."
- Line 355: "Note that some delineations (on the bottom left hand corner) can be noticed, which results from the threshold-based procedure to model the probabilistic distributions (see Sect. 3.3)."

First of all, I would expect that the thresholds values you provided would match the delineations seen in the figure, which from what I can see in Figure 3, are around 4 m, 13 m/s and 2.3 m for Hs, U and SWL respectively.

The simulated events seem to be quite dense above these thresholds (i.e., extremes events), while on the opposite side there are not that many events below the threshold. In essence, you are simulating more extreme than mild events, which I would not expect to be the case. Please explain if I am missing something here. This would mean that your probabilistic analysis for the flood volume is affected as well.

I have mentioned this issue in my previous review as well, but I did not recieve any response from the authors on this.

We thank Reviewer 3 for these new comments. In the light of this new analysis, we now have a better understanding of the issue raised by Reviewer 3 in the first round of review. We apologise for only partially addressing this issue. We have rechecked the different steps of the Monte Carlo procedure and have come to the same conclusion as Reviewer 3, namely that the random samples below the dependence threshold misrepresent the bulk of the multivariate distribution.

Three remedies to this problem have been applied:

1. To generate the random samples following Heffernan and Tawn (2004)'s approach, we now use the R package *texmex* (https://cran.r-project.org/web/packages/texmex/index.html) that is more stable than our in-house implementation. A clear improvement of stability was noticed when using Laplace instead of the Gumbel margins (see Keef et al., 2013);

2. We also notice that the problem particularly arises when the POT threshold is chosen too high. We propose to minimize the subjectivity in this selection by using the automatic threshold selection procedure developed by Northrop et al. (2017). This yielded lower values than the ones in the original version of the study, namely  $u_{\text{Hs}}$ =3.59m,  $u_{\text{SWL}}$ =2.37m, and  $u_{\text{U}}$ =9.51m/s;

3. In the original version of the study, we distinguished the skew surges and the tide. Given the long time series (~100 years of hindcast data), the difference with the direct approach (i.e. directly using the total water level) was reasonably small and we focus on this latter approach. The advantage is now to avoid a rescaling of the generated samples i.e. to avoid the delineations on the *SWL-Hs/U* plot, which were confusing.

New figure 3 shows the new randomly generated samples (with modifications of the colors to better indicate the different datasets as suggested by Reviewer #3).

New Figure 3: Overview of the N=50,000 randomly generated samples of offshore conditions (red dots). Blue dots correspond to the hindcast conditions used to fit the statistical methods described in Sect. 3.3. Green dots and squares correspond to the n=144 training data used to set-up the GP metamodel (the selection approach is detailed in Sect. 3.2). The squares correspond to cases that are deliberately selected outside the range of the red dots to cover a broader range of situations.

We added in Supplementary Material B diagnostic plots to confirm the goodness of the GPD fit. In addition, we provide in Supplementary Materials B the quantile-quantile plots to confirm that the generated samples adequately reproduce the marginals (see below Fig. S2-S5). We also updated the dependence measures' analysis (Supplementary Materials A).

On this basis, the whole global sensitivity analysis was re-conducted.

- As expected, the major change was a reduction in the flooding probability value (new Fig. 7) which is clearly consistent with Reviewer #3's analysis;
- The new sensitivity analysis (new Fig. 8) reveals that our previous conclusions remain quasi-similar with differences mainly corresponding to Shapley effects' changes of approximately of +/- 10% for the major sources of uncertainty (*SLR* and *SWL*) and a steeper time evolution;
- The new analysis of robustness (Sect. 5) still outlines the key role of the DEM but differs in the very long term (by 2200) with a lower influence of the RCP scenario. Supplementary materials E was also updated with these new results.

It should be mentioned that whatever efforts are made to tune the extreme value analysis, residual uncertainties will remain, mainly due to the sensitivity to the threshold selection (POT and dependence). This problem is clearly underlined in the discussion section (Sect. 5.2), and we believe that incorporating such uncertainties is a valuable line for future research.

Finally, although the extreme value analysis is an ingredient of the study (mainly based on previously published papers), it is the Shapley effect combined with Gaussian process metamodels that is new. This is the main message of our manuscript (see the abstract). We believe that the performance analysis (see below) as well as the physical significance of the sensitivity analysis (Sect. 4.3, Sect. 5.1 and Supplementary Materials E) give confidence in the value of this approach from an operational viewpoint.